# Identification of a *Kingella kingae* factor H binding protein that is the major determinant of serum resistance

**Kevin A. Hernandez**[1,2], **Eric A. Porsch**[2], **Vanessa L. Muñoz**[2], **Joseph W. St. Geme III**[1,2]*

**1** University of Pennsylvania Perelman School of Medicine, Philadelphia, Pennsylvania, United States of America, **2** Department of Pediatrics, Children's Hospital of Philadelphia, Philadelphia, Pennsylvania, United States of America

* stgemeiiij@chop.edu

## Abstract

*Kingella kingae* is a Gram-negative bacterium that has emerged as a leading cause of invasive disease in children between 6 months and 4 years of age. *K. kingae* initiates infection by colonizing the oropharynx, then breaches the oropharyngeal epithelium, enters the bloodstream, and disseminates to distant sites to cause disease, including osteomyelitis, septic arthritis, and endocarditis. To survive in the bloodstream and disseminate to sites of invasive disease, *K. kingae* produces a polysaccharide capsule and an exopolysaccharide that inhibit opsonin deposition and mediate resistance to complement-mediated serum killing. However, elimination of these extracellular polysaccharides only partially reduces *K. kingae* survival in human serum, suggesting that additional factors contribute to serum resistance. In this study, we found that *K. kingae* binds human factor H (FH), a negative regulator of the alternative complement pathway. In experiments using rat serum as a source of complement, we observed that *K. kingae* was able to utilize human FH to resist killing. Introduction of exogenous human FH into the juvenile rat infection model of *K. kingae* disease enhanced virulence in vivo, demonstrating the importance of FH binding in the pathogenesis of disease. Far-western blot analysis identified a 37-kDa outer membrane protein designated KK02920 that was responsible for FH binding and enhanced virulence in vivo in the presence of human FH. Loss of KK02920 virtually abrogated serum resistance, indicating that KK02920 is the major determinant of *K. kingae* serum resistance. Additional analysis revealed the presence of KK02920 across a collection of serum-resistant invasive and carrier *K. kingae* isolates, all of which can utilize human FH to resist complement-mediated killing. This work demonstrates the importance of a complement-regulator binding protein as a major mechanism of serum resistance in an encapsulated organism.

**Data availability statement:** The data supporting the findings in this work are available at figshare.com at the address: https://doi.org/10.6084/m9.figshare.29660843.

**Funding:** This work was supported by the National Institute of Allergy and Infectious Diseases (www.niaid.nih.gov) under award 1R01AI121015 and 1R01AI172841 (both to J.W.S.) and the National Science Foundation Graduate Research Fellowship (www.nsfgrfp.org) under awards DGE-1845298 (2021-2023) and DGE-2236662 (2023-2024) (both to K.A.H.) and DGE-1321851 (to V.L.M.). The sponsors played no role in the study design, data collection and analysis, decision to publish, or preparation of the manuscript.

**Competing interests:** The authors have declared that no competing interests exists.

## Author summary

*Kingella kingae* is an emerging pediatric bacterial pathogen and is a major cause of a variety of invasive diseases in young children. Survival in the bloodstream is essential for *K. kingae* to cause invasive disease. In previous work, we established that *K. kingae* produces a polysaccharide capsule and an exopolysaccharide that mediate resistance to complement-mediated serum killing, an important component of the innate immune system. However, elimination of the capsule and the exopolysaccharide results in only a partial reduction in serum resistance. In this work, we discovered that *K. kingae* binds the factor H (FH) complement regulator, resulting in resistance to killing by rat complement and enhanced virulence in the rat model of invasive *K. kingae* disease. In addition, we identified a surface protein designated KK02920 that is responsible for FH binding and is the major determinant of *K. kingae* serum resistance. Further analysis revealed the uniform presence of KK02920 in a group of carrier and invasive isolates of *K. kingae*. Our study highlights that *K. kingae* employs multiple mechanisms to resist complement killing and survive in the bloodstream, a key aspect of *K. kingae* pathogenicity.

## Introduction

*Kingella kingae* is a Gram-negative, encapsulated coccobacillus and a member of the *Neisseriaceae* family. As a consequence of improved culture techniques and the increased use of molecular diagnostics over the last two decades, *K. kingae* has emerged as the most common etiological agent of osteomyelitis and septic arthritis in children 6–48 months of age in many countries. [1–3] *K. kingae* is also a common cause of bacteremia in young children and an important cause of endocarditis in children and adults. [2,4] *K. kingae* is a common member of the commensal oropharyngeal flora in children and must breach the oropharyngeal epithelium, enter the bloodstream, and disseminate to distant sites to cause disease. [5–8].

The bloodstream is a hostile environment for microorganisms, reflecting the presence of innate immune mechanisms that serve as an immediate defense against pathogenic bacteria and other microbes. A crucial component of the innate immune system is the complement system, a tightly regulated network of plasma and membrane-associated serum proteins. Proteolytic cleavage of these proteins results in a cascade reaction on the surface of pathogens to induce chemotaxis and opsonization for elimination by immune effector cells or to mediate direct killing by microbial lysis. [9,10] The complement system is activated via the classical, the lectin, or the alternative pathway. The classical and lectin pathways are initiated through distinct pathogen-binding proteins, while the alternative pathway has been widely considered to be initiated through spontaneous deposition of complement on the bacterial surface. [9,10] However, recent studies have shown that spontaneous complement deposition may be inefficient and that the alternative pathway may

serve primarily as an amplification mechanism, triggered by other means such as the classical and lectin pathways. [11–13] Despite the differences in initiation, these three pathways converge with the formation of a C3 convertase, cleavage of complement protein C3, and subsequent deposition of complement protein fragment C3b on the bacterial surface, marking the pathogen for elimination by immune effector cells and initiating the formation of the pore-forming membrane attack complex for bacterial lysis. [9,10] Once the complement system is initiated, the downstream domino-like cascade of complement cleavage events results in the release of inflammatory complement fragments and the recruitment of immune effector cells for bacterial clearance. To prevent spontaneous activation on host cells and tissue damage from continual inflammation, the complement system is highly regulated at a variety of steps through an assortment of effector proteins. [14]

In order for *K. kingae* to survive intravascularly, it must evade the innate immune system. We have previously demonstrated that the *K. kingae* polysaccharide capsule and galactan exopolysaccharide are crucial for resisting complement-mediated serum killing and neutrophil phagocytosis and killing and are required for full virulence in a juvenile rat infection model of *K. kingae* disease. [8,15] These polysaccharides contribute to high-level resistance to complement-mediated serum killing through inhibition of the classical complement pathway by impeding deposition of antibodies and complement fragments. [8] Although elimination of the capsule and exopolysaccharide results in a marked decrease in serum resistance, the capsule-deficient, exopolysaccharide-deficient mutant strain retains significant residual resistance, suggesting that other bacterial factors also influence *K. kingae* serum resistance. [8] Furthermore, *K. kingae* is fully resistant to human serum when only the alternative pathway is active, suggesting the presence of mechanisms of resistance to both the alternative pathway and the classical pathway. [8]

A number of bacterial pathogens have evolved mechanisms to exploit the regulatory arm of the complement pathway, including binding negative regulators of the complement pathway to downregulate complement activation on the bacterial surface. [16] The primary negative regulator of the alternative pathway is factor H (FH), which initiates dissociation of the alternative pathway C3 convertase (limiting C3b deposition on the bacterial surface) and recruits factors to inactivate C3b. [14,17–21]

In this study, we found that *K. kingae* can readily bind FH from human serum, with the capsule and exopolysaccharide hindering full FH binding. In addition, we observed that *K. kingae* binding of FH confers resistance to complement-mediated killing and enhanced virulence in juvenile rats. Further, we identified a *K. kingae* outer membrane protein called KK02920 that is responsible for binding human FH and for mediating resistance to complement-mediated serum killing and enhanced virulence in the juvenile rat model of invasive *K. kingae* disease.

## Results

### *K. kingae* binds FH and/or FHL-1 from human serum

In considering mechanisms of resistance to the alternative complement pathway that may contribute to the intermediate serum survival profile of *K. kingae* when the polysaccharide capsule and exopolysaccharide are absent, we sought to determine if *K. kingae* binds FH, the primary soluble negative regulator of the alternative pathway. To quantitate levels of FH deposition on the surface of *K. kingae*, we performed flow cytometry with *K. kingae* strain KK01, a naturally occurring nonspreading and noncorroding variant of septic arthritis clinical isolate 269–492. [7] We used heat-inactivated normal human serum (HI-NHS) as a source of FH, aiming to inactivate the bactericidal effects of complement while retaining FH stability. [22] An inoculum of $2 \times 10^8$ CFU of strain KK01 was incubated with HI-NHS concentrations of 0%, 10%, and 50%, using *E. coli* strain DH5α as a negative control for FH binding. We used the monoclonal antibody OX-24, which detects FH and the FH alternative splice variant, FH-like protein 1 (FHL-1). [23–25] Compared to DH5α, there was a marked increase in signal with OX-24, indicating significant FH and/or FHL-1 (FH/FHL-1) deposition on the surface of strain KK01 when incubated with 50% HI-NHS (Fig 1).

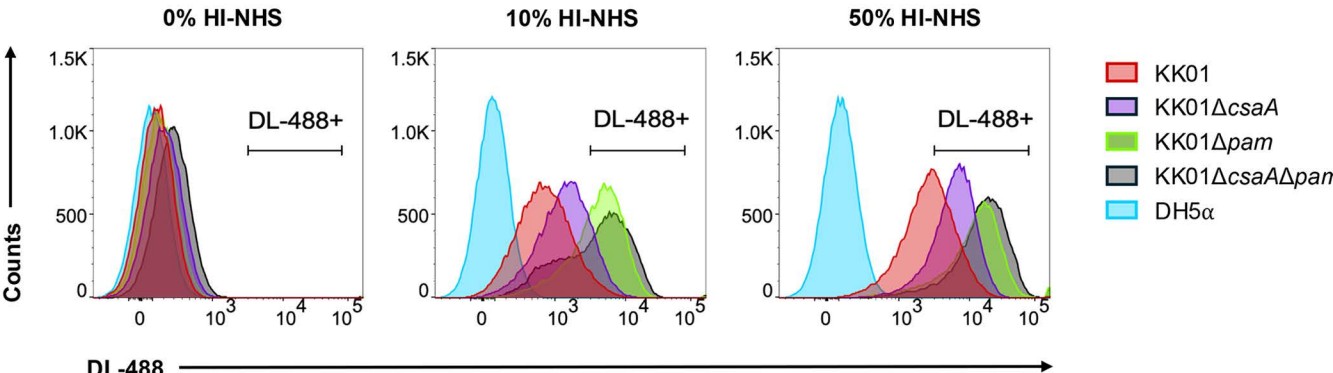

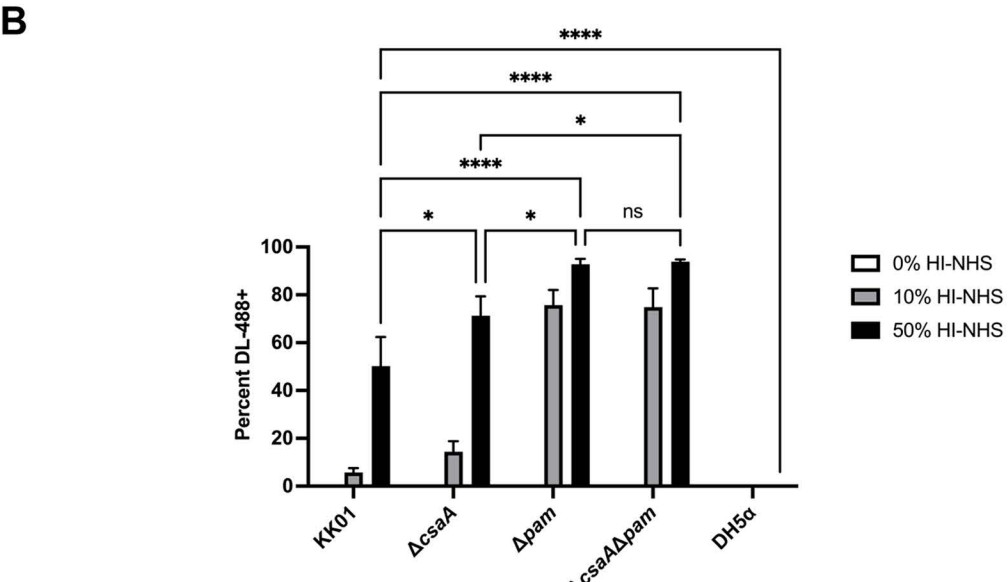

**Fig 1. *K. kingae* binds FH and/or FHL-1 from human serum.** Binding of FH and/or FHL-1 to the bacterial surface of *K. kingae* strains KK01, KK01Δ*csaA*, KK01Δ*pam*, and KK01Δ*csaA*Δ*pam* and *E. coli* strain DH5α was determined using flow cytometry. Bacteria were incubated with 10% or 50% HI-NHS for 1 h or with PBS (0% HI-NHS) as a primary and secondary antibody-only control. Cells were stained with propidium iodine (PI) prior to analysis; 50,000 events per biological replicate were analyzed. A total of three biological replicates were performed (*n* = 3). (A) Representative histograms are shown. Histograms: KK01, red; KK01Δ*csaA*, purple; KK01Δ*pam*, green; KK01Δ*csaA*Δ*pam*, black; DH5α, blue. Cells were gated for PI+ and DyLight-488+ (DL-488+) cells. One representative experiment out of three performed is shown. (B) Quantified data from panel A. The bars from the 0% HI-NHS group are negligible in size due to low signal. The bars from all serum concentrations for strain DH5α are negligible in size due to low signal. Data are presented as means, and the error bars represent the standard error of the mean. Statistical significance was determined by 2-way analysis of variance (ANOVA) with Tukey's correction for multiple comparisons. *, $P < 0.05$; ****, $P < 0.0001$.

We have previously demonstrated that the presence of either the *K. kingae* capsule or the exopolysaccharide blocks complement and antibody opsonization. [8,15] To determine the effect of the capsule and exopolysaccharide on FH/FHL-1 binding, we performed flow cytometry with *K. kingae* strains KK01Δ*csaA* (a capsule-deficient mutant lacking the *csaA* capsule synthesis gene), [26] KK01Δ*pam* (an exopolysaccharide-deficient mutant lacking the *pamABCDE* galactan exopolysaccharide synthesis operon), [27] and KK01Δ*csaA*Δ*pam* (a capsule- and exopolysaccharide-deficient mutant lacking

the *csaA* capsule synthesis gene and the *pamABCDE* galactan exopolysaccharide synthesis operon) [26] after incubation with 50% HI-NHS. Elimination of the capsule resulted in a significant increase in FH/FHL-1 binding compared to strain KK01 (Fig 1). Elimination of the exopolysaccharide resulted in an even greater increase in FH/FHL-1 binding compared to strain KK01, similar to strain KK01Δ*csaA*Δ*pam* lacking both capsule and exopolysaccharide. These data demonstrate that *K. kingae* can bind FH/FHL-1 in human serum and that the capsule and exopolysaccharide impede FH/FHL-1 binding.

### *K. kingae* binding of human FH results in resistance to complement-mediated killing

In order to assess whether *K. kingae* binding of FH results in serum resistance, we used rat serum as a source of active complement and incubated *K. kingae* strains KK01 and KK01Δ*csaA*Δ*pam* for 1 h with either normal rat serum (NRS) or heat-inactivated NRS (HI-NRS) concentrations ranging from 1% to 25% as a source of inactive complement. Serum sensitivity was calculated by dividing the number of colony forming units (CFU) recovered from NRS by the number of CFU recovered from HI-NRS. As shown in Fig 2A and 2B, survival of strains KK01 and KK01Δ*csaA*Δ*pam* decreased markedly with increasing concentrations of NRS, with no survival at a NRS concentration of 5% or greater, indicating that strains KK01 and KK01Δ*csaA*Δ*pam* are susceptible to the bactericidal activity of rat complement.

To examine whether binding of human FH, independent of FHL-1, results in resistance to complement-mediated killing, we supplemented each concentration of rat serum with 100 µg/mL purified human FH. As shown in Fig 2A and 2B, addition of human FH restored survival of strains KK01 and KK01Δ*csaA*Δ*pam* at all serum concentrations. The addition of human FH did not affect the bactericidal activity of rat complement, as evidenced by killing of DH5α (S1 Fig).

The concentration of FH in normal human serum is typically approximately 500 µg/mL, although levels can range between 116 and 562 µg/mL depending on genetic and environmental factors. [28] The levels of FH in serum are lowest in neonates and increase with age. [28–30] Given the wide range of FH concentrations in the human population, we sought to determine the lowest concentration of human FH that promotes substantial survival of strain KK01 in NRS and performed serum bactericidal assays using 5% NRS with FH concentrations ranging from 0 µg/mL to 100 µg/mL. As shown

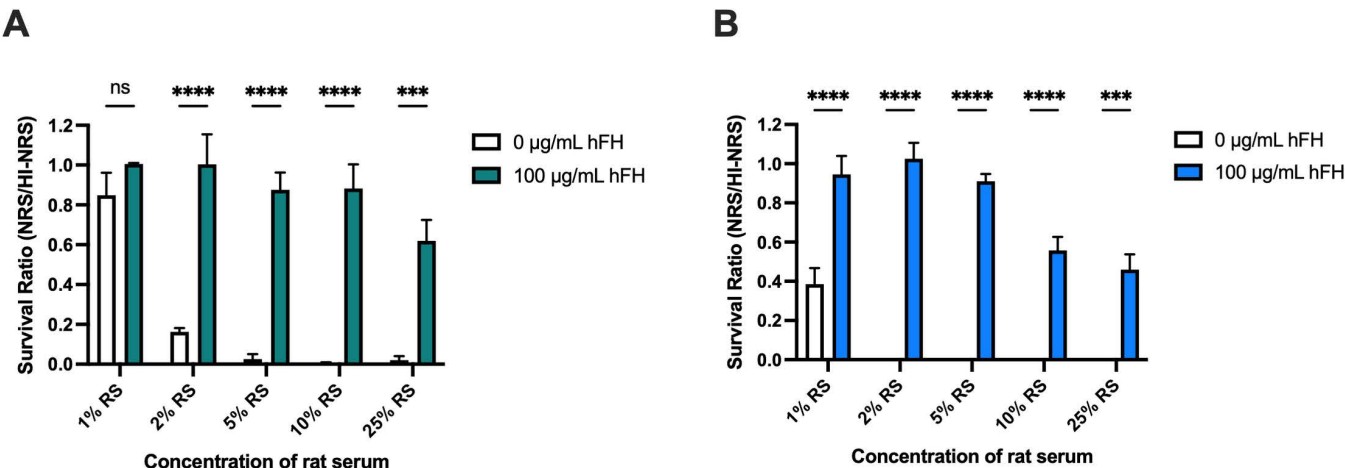

**Fig 2. *K. kingae* binds human FH to resist rat complement-mediated killing.** *K. kingae* strains (A) KK01 and (B) KK01Δ*csaA*Δ*pam* (~$10^3$ CFU) were incubated with 1%, 2%, 5%, 10%, or 25% normal rat serum (NRS) or heat-inactivated NRS (HI-NRS) with either 0 µg/mL or 100 µg/mL of human factor H (hFH) for 1 h. (A – B) The survival ratio was calculated by dividing NRS CFU counts by the HI-NRS CFU counts. A total of 3 biological replicates were performed (*n* = 3). The survival ratios are negligible for strain KK01 incubated with 0 µg/mL hFH and rat serum concentrations above 2% and for strain KK01Δ*csaA*Δ*pam* incubated with 0 µg/mL hFH and rat serum concentrations above 1%. Data are presented as means, and the error bars represent the standard error of the mean. Statistical significance was determined by 2-way analysis of variance (ANOVA) with Sidak's correction for multiple comparisons. ***, $P < 0.0005$; ****, $P < 0.0001$.

in S2 Fig, as little as 1 µg/mL of human FH was sufficient to restore survival of strain KK01 to a survival ratio of approximately 0.8. These results suggest that *K. kingae* can bind human FH to resist complement-mediated killing and that *K. kingae* requires minimal concentrations of human FH to achieve resistance, consistent with physiological levels in human serum.

Given the ability of *K. kingae* to utilize minimal concentrations of human FH to resist the bactericidal effects of rat complement, we sought to confirm that the three complement pathways were active in the rat serum. Calcium is required for the classical and lectin complement pathways, and magnesium is necessary for the alternative complement pathway. [31] In serum assays, EGTA chelates $Ca^{2+}$, blocking activation of the classical and lectin pathways, and the concomitant addition of $Mg^{2+}$ maintains the alternative pathway. [31] We performed serum bactericidal assays using 5% NRS with the addition of EGTA and $Mg^{2+}$. As shown in S3 Fig, the addition of EGTA resulted in nearly full restoration of strain KK01 survival, and the addition of both EGTA and $Mg^{2+}$ resulted in a survival ratio of about 0.65. These results suggest that the classical and/or lectin pathway as well as the alternative pathway contribute to the killing of *K. kingae* in rat serum.

### Treatment with human FH results in enhanced *K. kingae* virulence in vivo

Given the ability of *K. kingae* to use human FH to mediate resistance to rat complement-mediated lysis in vitro, we sought to address whether *K. kingae* binding of human FH affects virulence in vivo using an established juvenile rat infection model of *K. kingae* disease. [8,32,33] Five-day-old Sprague-Dawley rat pups were inoculated intraperitoneally with 1 x $10^7$ CFU of strain KK01 or strain KK01Δ*csaA*Δ*pam* supplemented with either 50 µg of human FH in PBS or PBS alone. Mock infected rat pups were inoculated with 50 µg human FH alone as a control. Among the rat pups inoculated with human FH alone, 100% of animals survived at 120 hours post-infection (Fig 3), indicating that introduction of exogenous human FH in this animal model does not result in mortality in the absence of *K. kingae*. Among the rat pups inoculated with strain KK01 with PBS, 100% of animals survived at 120 hours post-infection (Fig 3). In contrast, 23.5% of animals inoculated with strain KK01 incubated with human FH survived ($P<0.0001$). Among the rat pups inoculated with strain KK01Δ*csaA*Δ*pam* with PBS, 100% of animals survived at 120 hours-post infection (Fig 3). In contrast, 33.3% of animals inoculated with strain KK01Δ*csaA*Δ*pam* incubated with human FH survived ($P<0.0001$). These results suggest that *K. kingae* can use exogenous human FH to evade the rat immune system in vivo, consistent with our in vitro results.

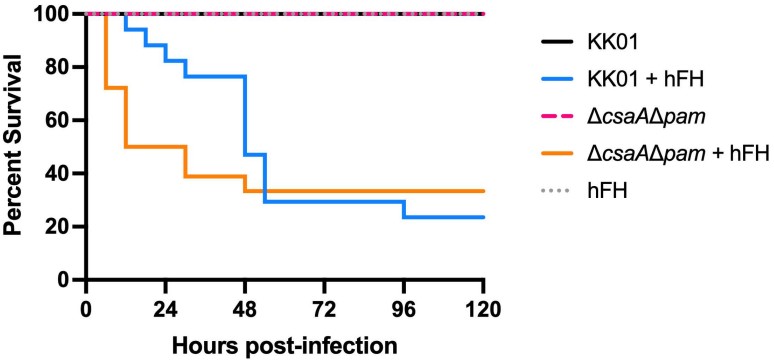

**Fig 3. Treatment with human FH results in enhanced *K. kingae* virulence in infant rats.** The graph plots Kaplan-Meier survival curves for five-day-old Sprague-Dawley rats inoculated via the intraperitoneal (i.p.) route with either 50 µg human FH (hFH) in 0.15mL PBS, 1 x $10^7$ CFU of *K. kingae* strain KK01 in PBS, 1 x $10^7$ CFU of KK01 with 50 µg hFH, 1 x $10^7$ CFU of KK01Δ*csaA*Δ*pam* in PBS, or 1 x $10^7$ CFU of KK01Δ*csaA*Δ*pam* with 50 µg hFH. Data are for 16 animals inoculated with hFH, 17 animals inoculated with KK01 or KK01 with 50 µg hFH, and 18 animals inoculated with KK01Δ*csaA*Δ*pam* or KK01Δ*csaA*Δ*pam* with hFH. There was no mortality in the following cohorts: KK01, KK01Δ*csaA*Δ*pam*, and hFH. Statistical significance was determined by Log-rank (Mantel-Cox) test: KK01 & KK01+hFH (****, $P<0.0001$); KK01Δ*csaA*Δ*pam* & KK01Δ*csaA*Δ*pam*+hFH (****, $P<0.0001$); KK01+hFH & KK01Δ*csaA*Δ*pam*+hFH (ns, $P>0.05$); KK01 & hFH (ns, $P>0.05$).

## A *K. kingae* outer membrane protein binds human FH

Some pathogenic bacteria have acquired the ability to hijack human FH, and in Gram-negative bacteria, the mechanism of FH binding is commonly through a specific FH binding integral outer membrane protein. [34] To identify FH binding proteins in the *K. kingae* outer membrane, outer membrane protein preparations of *K. kingae* strains KK01 and KK01Δ*csaA*Δ*pam* were resolved by SDS-PAGE and subjected to far-western blot analysis, probing with purified human FH and using DH5α and bovine serum albumin (BSA) as controls. Using this approach, we identified a ~37-kDa band in both strain KK01 and strain KK01Δ*csaA*Δ*pam* (Fig 4).

To identify *K. kingae* outer membrane proteins migrating at 37-kDa, an SDS-PAGE gel was repeated and stained with Coomassie blue (Fig 4). The 37-kDa band from strain KK01Δ*csaA*Δ*pam* was excised and subjected to mass spectrometry analysis. Top results identified by mass spectrometry were filtered by: abundance in the sample, amino acid sequence similarity to any known FH binding proteins through NCBI BLAST, FH binding annotations through UniProt, and structural similarities to known FH binding proteins through HHpred. This analysis identified two uncharacterized lipoproteins designated KK02010 and KK02920. Both KK02010 and KK02920 were annotated as *K. kingae* factor H binding protein domain-containing proteins (NCBI Reference Sequence: WP_019389828.1 and WP_166503244.1, respectively), in reference to the factor H-binding protein (FHbp) of *N. meningitidis*. Based on analysis using NCBI BLAST, the protein outside of the genus *Kingella* with greatest homology to KK02010 is a factor H binding protein domain-containing protein in *N. cinerea* (NCBI Reference Sequence: WP_308032693.1), with 42% identity to KK02010. The protein outside of the genus *Kingella* with greatest homology to KK02920 is a different factor H binding protein domain-containing protein in *N. cinerea* (NCBI Reference Sequence: WP_314366668.1), with 31.62% identity to KK02920. Annotations in UniProt of KK02010 (F5S9U2) and KK02920 (F5S9U2) revealed that these proteins contain factor H binding protein-like C terminal domains. Amino acid sequence analyses using HHpred (PDB_mmCIF70_3_Jan database) revealed predicted protein structures similar to *N. meningitidis* FHbp for both KK02010 and KK02920.

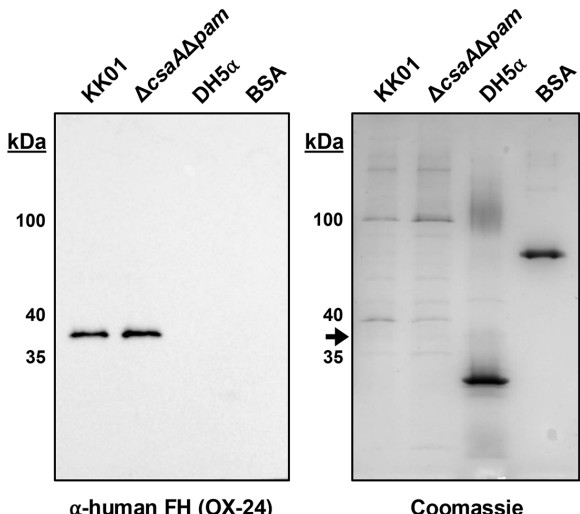

**Fig 4. A *K. kingae* outer membrane protein binds human FH.** Outer membrane preparations were isolated from *K. kingae* strains KK01 and KK01Δ*csaA*Δ*pam* and *E. coli* strain DH5α. These outer membrane preparations were separated on a 12.5% SDS-PAGE gel and transferred to a nitrocellulose membrane or stained with Coomassie blue. A far-western blot was performed by incubating the membrane with 100 μg/mL purified human FH, anti-human FH mAb (OX-24), and an HRP-coupled anti-mouse IgG. One band of ~37-kDa was present for KK01 and KK01Δ*csaA*Δ*pam* in the far-western blot. DH5α and bovine serum albumin (BSA) served as negative binding controls. The black arrow next to the Coomassie-stained gel represents the ~37-kDa protein band that binds human FH. Representative images are shown.

## Outer membrane protein KK02920 is responsible for *K. kingae* binding of human FH

To assess the possible role of the KK02010 and KK02920 proteins in binding FH, we generated strain KK01 and strain KK01Δ*csaA*Δ*pam* mutants lacking these proteins (KK01Δ*02010*, KK01Δ*02920*, KK01Δ*csaA*Δ*pam*Δ*02010,* and KK01Δ*csaA*Δ*pam*Δ*02920*). The mutants lacking KK02010 retained the ability to bind FH as assessed by flow cytometry (Fig 5A & B) and far-western blot analysis (Fig 5C). In contrast, the mutants lacking KK02920 were completely incapable of binding FH (Fig 5). Complementation of the *02920* deletion at a separate locus on the chromosome [KK01Δ*02920*(*02920*) and KK01Δ*csaA*Δ*pam*Δ*02920*(*02920*)] restored FH binding (Fig 5). These results indicate that KK02920 is an outer membrane protein that binds FH and is the sole factor responsible for FH binding in strain KK01.

## Enhanced *K. kingae* virulence in the presence of human FH is dependent on KK02920

We next sought to determine whether binding of human FH via KK02920 is responsible for the enhanced virulence of *K. kingae* when treated with human FH prior to infection. Five-day-old Sprague-Dawley rat pups were inoculated intra-peritoneally with $1 \times 10^7$ CFU of strain KK01Δ*csaA*Δ*pam*, KK01Δ*csaA*Δ*pam*Δ*02920*, or KK01Δ*csaA*Δ*pam*Δ*02920*(*02920*) supplemented with either 50 µg of human FH in PBS or PBS alone (Fig 6). Among the rat pups inoculated with strain KK01Δ*csaA*Δ*pam* with PBS, 100% of animals survived at 120 hours post-infection. In contrast, only 22.2% of animals inoculated with strain KK01Δ*csaA*Δ*pam* incubated with human FH survived ($P<0.0001$). Among the rat pups inoculated with strain KK01Δ*csaA*Δ*pam*Δ*02920* with PBS, 100% of animals survived at 120 hours post-infection. Interestingly, 94.4% of animals inoculated with strain KK01Δ*csaA*Δ*pam*Δ*02920* incubated with human FH survived ($P>0.05$). Among the rat pups inoculated with strain KK01Δ*csaA*Δ*pam*Δ*02920*(*02920*) with PBS, 94.4% of animals survived at 120 hours post-infection. In contrast, 50.0% of animals inoculated with strain KK01Δ*csaA*Δ*pam*Δ*02920*(*02920*) incubated with human FH survived ($P<0.005$). These results support the conclusion that KK02920 is necessary for *K. kingae* to bind human FH to evade the rat immune system in vivo.

## KK02920 plays a major role in inhibiting complement-mediated serum killing of *K. kingae* by resisting the alternative complement pathway

To extend the observation that the KK02920 mutants are unable to bind FH, we examined the contribution of KK02920 to serum resistance relative to the polysaccharide capsule and the exopolysaccharide. We hypothesized that in the absence of the capsule and exopolysaccharide, *K. kingae* can resist human serum killing by binding human FH via KK02920 to inhibit complement deposition on the bacterial surface. At an inoculum of $1.0 \times 10^6$ CFU, *K. kingae* strains KK01, KK01Δ*02920*, KK01Δ*02920*(*02920*), KK01Δ*csaA*Δ*pam*, KK01Δ*csaA*Δ*pam*Δ*02920*, and KK01Δ*csaA*Δ*pam*Δ*02920*(*02920*), along with serum-susceptible *E. coli* strain DH5α were incubated with 50% human serum for 1h. As shown in Fig 7A, strains KK01Δ*02920* and KK01Δ*csaA*Δ*pam*Δ*02920* displayed a 3-log decrease in serum survival compared to their parent strains, comparable to the result with strain DH5α, indicating complete abrogation of serum resistance. Complementation of both KK01Δ*02920*(*02920*) and KK01Δ*csaA*Δ*pam*Δ*02920*(*02920*) with wild type *02920* restored serum resistance.

To confirm that the disruption of *02920* in KK01Δ*02920* did not affect capsule or exopolysaccharide production, we extracted capsule and exopolysaccharide from strains KK01, KK01Δ*02920*, KK01Δ*csaA*Δ*pam*, and KK01Δ*csaA*Δ*pam*Δ*02920*. As shown in S4A Fig, the levels of capsule were similar in strains KK01 and KK01Δ*02920*, as visualized with alcian blue staining. As shown in S4B Fig, the levels of exopolysaccharide were also similar in strains KK01 and KK01Δ*02920*, as visualized by western blot with an anti-galactan exopolysaccharide antiserum. Together these results indicate that KK02920 is a major factor responsible for resistance to complement-mediated killing, with contributions from the *K. kingae* capsule and exopolysaccharide.

Given that KK02920 is required for binding human FH, we investigated its role in evading the alterative complement pathway during serum-mediated killing. We performed serum bactericidal assays using EGTA and $Mg^{2+}$ to selectively

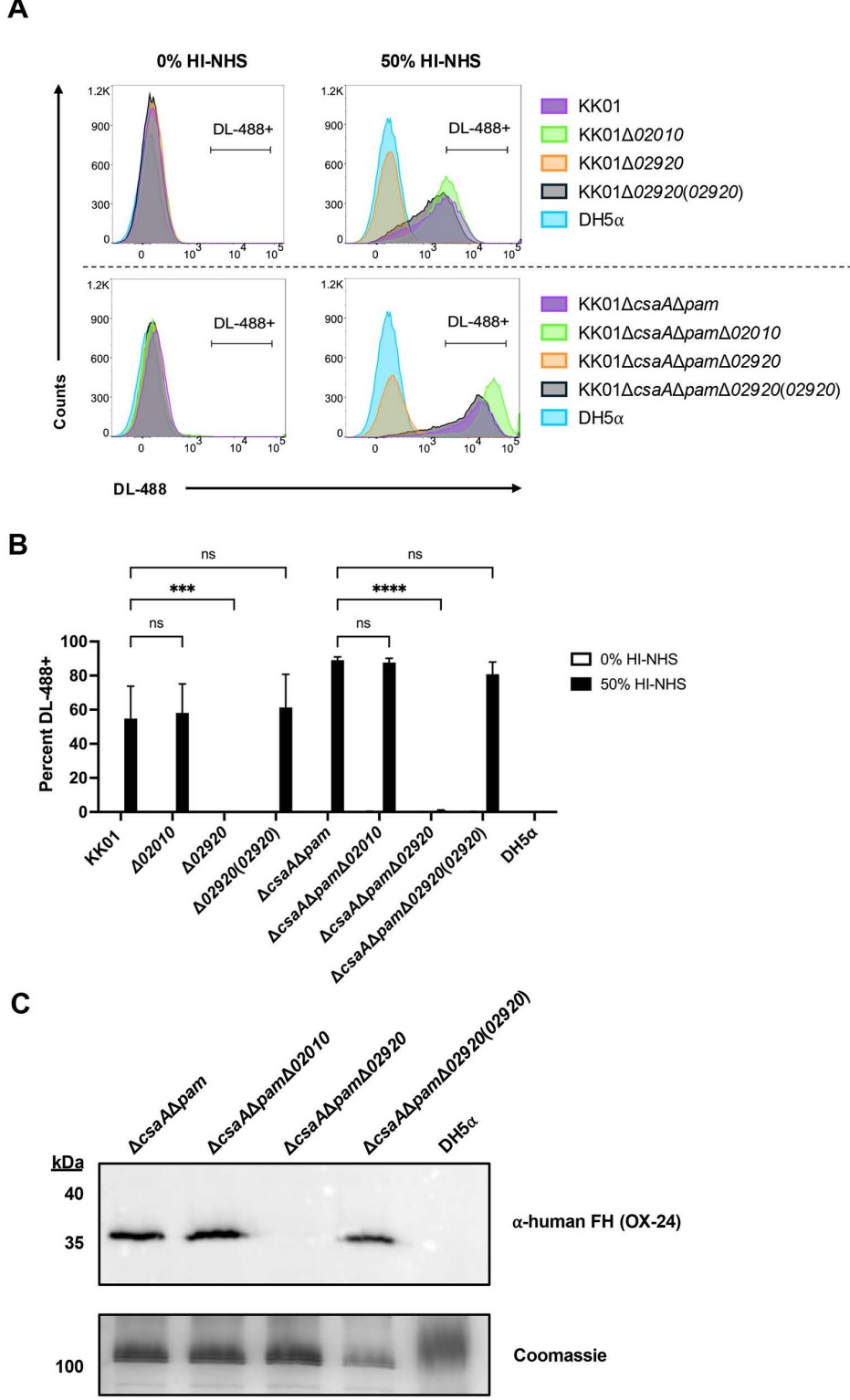

**Fig 5. Outer membrane protein KK02920 is responsible for *K. kingae* binding of human FH.** (A) Binding of factor H to the bacterial surface of *K. kingae* strains KK01, KK01Δ*02010*, KK01Δ*02920*, KK01Δ*02920*(*02920*), KK01Δ*csaA*Δ*pam*, KK01Δ*csaA*Δ*pam*Δ*02010*, KK01Δ*csaA*Δ*pam*Δ*02920*, and KK01Δ*csaA*Δ*pam*Δ*02920*(*02920*) and *E. coli* strain DH5α was determined using flow cytometry. Bacteria were incubated with 50% HI-NHS for 1 h or

with PBS (0% HI-NHS) as a primary and secondary antibody-only control. Cells were stained with propidium iodine (PI) prior to analysis; 50,000 events per biological replicate were analyzed. The gating strategy demonstrated in Fig 1 is the exact gating strategy used here. Representative histograms are shown: KK01 and KK01ΔcsaAΔpam, purple; KK01Δ02010 and KK01ΔcsaAΔpamΔ02010, green; KK01Δ02920 and KK01ΔcsaAΔpamΔ02920, orange; KK01Δ02920(02920) and KK01ΔcsaAΔpamΔ02920(02920), black; DH5α, blue. Top left histogram: strain KK01 background treated with 0% HI-NHS; bottom left histogram: strain KK01ΔcsaAΔpam background treated with 0% HI-NHS; top right histogram: strain KK01 background treated with 50% HI-NHS; bottom right histogram: strain KK01ΔcsaAΔpam background treated with 50% HI-NHS. Cells were gated for PI+ and DyLight-488+ (DL-488+) cells. One representative experiment out of three performed is shown (n = 3). (B) Quantified data from panel A. The bars from the 0% HI-NHS group are negligible in size due to low signal. The bars from both serum concentrations for strain DH5α are negligible in size due to low signal. The percentages represent events that registered as DyLight 488 positive (DL-488+). A total of three biological replicates were performed (n = 3). The bars from the 0% HI-NHS group are negligible in size due to low signal. Data are presented as means, and the error bars represent the standard error of the mean. Statistical significance was determined by 2-way analysis of variance (ANOVA) with Tukey's correction for multiple comparisons. ***, $P < 0.001$; ****, $P < 0.0001$. (C) Outer membrane preparations were isolated from *K. kingae* strains KK01ΔcsaAΔpam, KK01ΔcsaAΔpamΔ02010, KK01ΔcsaAΔpamΔ02920, and KK01ΔcsaAΔpamΔ02920(02920) and *E. coli* strain DH5α. These outer membrane preparations were separated on a 12.5% SDS-PAGE gel and transferred to a nitrocellulose membrane or stained with Coomassie blue. A far-western blot was performed by incubating the membrane in 10% HI-NHS, anti-human FH mAb (OX-24), and an HRP-coupled anti-mouse IgG. The ~ 37-kDa band represents the outer membrane protein that binds human FH. The ~ 100-kDa band represents a loading control. A representative image is shown.

inhibit the classical and lectin pathways, thereby isolating the alternative pathway. As shown in Fig 7B, the addition of EGTA and Mg²⁺ restored survival of strain KK01ΔcsaAΔpam to the level of strain KK01. In contrast, the addition of EGTA and Mg²⁺ to serum resulted in survival of strain KK01Δ02920 that was reduced by ~80% compared to strain KK01, indicating that KK02920 affects resistance to the alternative pathway. Similarly, the addition of EGTA and Mg²⁺ to serum resulted in survival of strain KK01ΔcsaAΔpamΔ02920 that was reduced by >99% compared to strain KK01ΔcsaAΔpam, again indicating that KK02920 affects resistance to the alternative pathway. Although not statistically significant (P = 0.19), the addition of EGTA and Mg²⁺ to serum resulted in survival of strain KK01ΔcsaAΔpamΔ02920 that was reduced by ~20% compared to strain KK01Δ02920. These results suggest that KK02920 plays a critical role in protecting against the alternative pathway in serum killing, with a modest contribution from the capsule and exopolysaccharide.

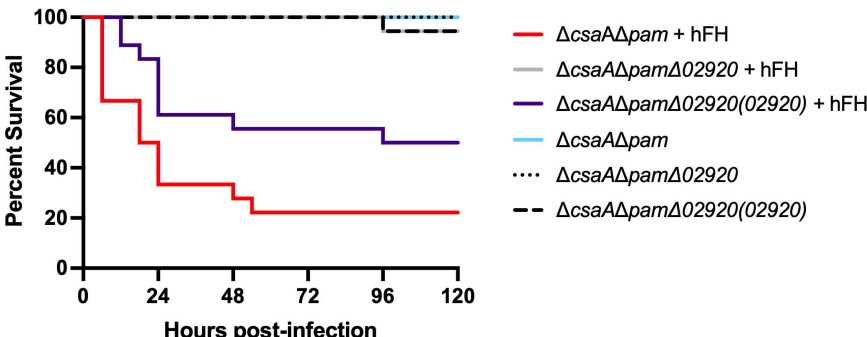

**Fig 6. Enhanced *K. kingae* virulence in the presence of human FH is dependent on KK02920.** The graph plots Kaplan-Meier survival curves of five-day-old Sprague-Dawley rats inoculated via the intraperitoneal (i.p.) route with either 1 x 10⁷ CFU of *K. kingae* strain KK01ΔcsaAΔpam in PBS, 1 x 10⁷ CFU of KK01ΔcsaAΔpam with 50 μg human factor H (hFH), 1 x 10⁷ CFU of KK01ΔcsaAΔpamΔ02920 in PBS, 1 x 10⁷ CFU of KK01ΔcsaAΔpamΔ02920 with 50 μg hFH, 1 x 10⁷ CFU of KK01ΔcsaAΔpamΔ02920(02920) in PBS, or 1 x 10⁷ CFU of KK01ΔcsaAΔpamΔ02920(02920) with 50 μg hFH. Data are for 16 animals inoculated with KK01ΔcsaAΔpam hFH or 18 animals for all other cohorts. Statistical significance was determined by Log-rank (Mantel-Cox) test: KK01ΔcsaAΔpam & KK01ΔcsaAΔpam+hFH (****, $P < 0.0001$); KK01ΔcsaAΔpamΔ02920 & KK01ΔcsaAΔpamΔ02920+hFH (not significant, $P > 0.05$); KK01ΔcsaAΔpamΔ02920(02920) & KK01ΔcsaAΔpamΔ02920(02920) + hFH (**, $P < 0.005$); KK01ΔcsaAΔpam+hFH & KK01ΔcsaAΔpamΔ02920+hFH (****, $P < 0.0001$); KK01ΔcsaAΔpamΔ02920+hFH & KK01ΔcsaAΔpamΔ02920(02920) (**, $P < 0.005$); KK01ΔcsaAΔpam+hFH & KK01ΔcsaAΔpamΔ02920(02920) + hFH (*, $P < 0.05$).

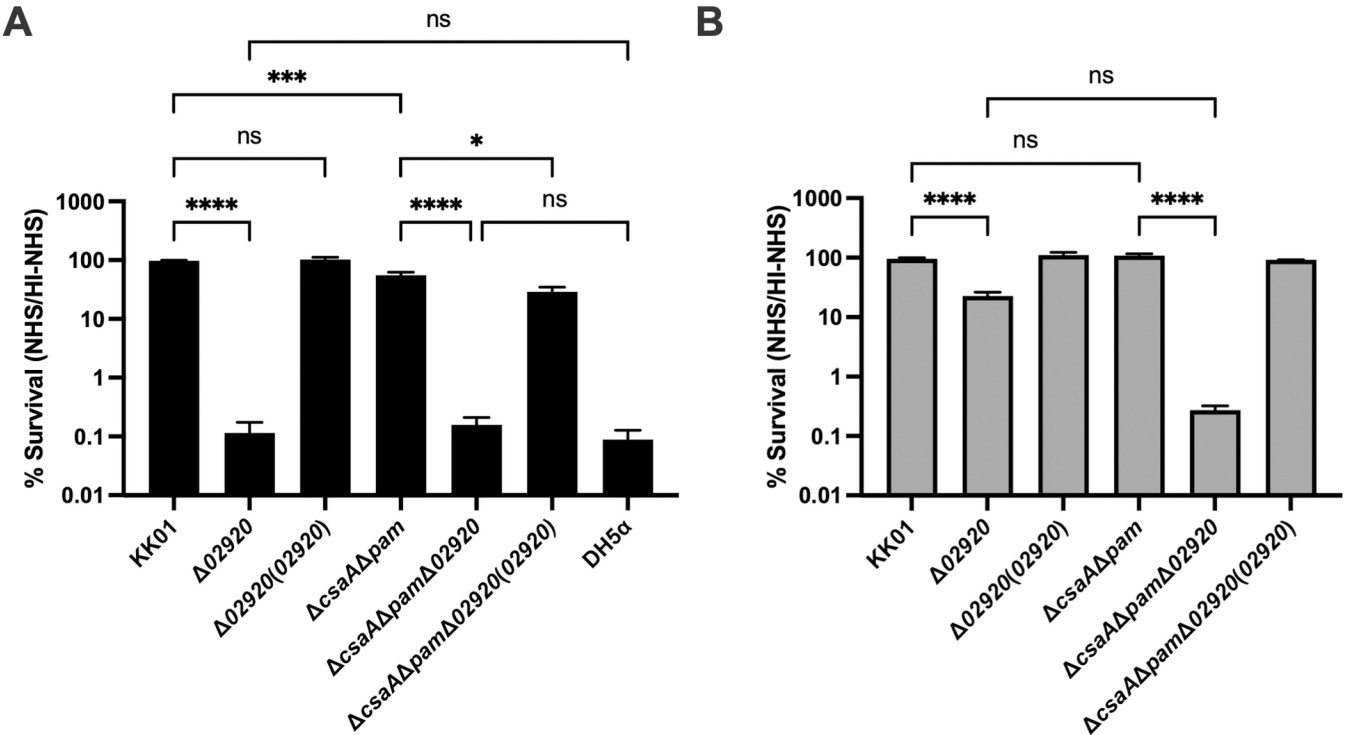

**Fig 7. KK02920 plays a major role in inhibiting complement-mediated serum killing of *K. kingae* by resisting the alternative complement pathway.** (A) *K. kingae* strains KK01, KK01Δ*02920*, KK01 Δ*02920*(*02920*), KK01Δ*csaA*Δ*pam*, KK01Δ*csaA*Δ*pam*Δ*02920*, KK01Δ*csaA*Δ*pam*Δ*02920*(*02920*), and *E. coli* strain DH5α (~$10^6$ CFU) were incubated with 50% NHS for 1 h. (B) *K. kingae* strains KK01, KK01Δ*02920*, KK01Δ*02920*(*02920*), KK01Δ*csaA*Δ*pam*, KK01Δ*csaA*Δ*pam*Δ*02920*, and KK01Δ*csaA*Δ*pam*Δ*02920*(*02920*) (~$10^6$ CFU) were incubated with 50% NHS plus 5mM EGTA and 9mM $Mg^{2+}$ for 1 h. The percent (%) survival was calculated by dividing the NHS CFU counts by the HI-NHS CFU counts and multiplying by 100. A total of 3 biological replicates were performed (*n*=3). Data are presented as means, and the error bars represent the standard error of the mean. Statistical significance was determined by 1-way analysis of variance (ANOVA) with Tukey's correction for multiple comparisons. *, $P<0.05$; ***, $P<0.001$; ****, $P<0.0001$.

### FH retains cofactor activity when bound to *K. kingae*, and this activity is dependent on KK02920

FH regulates the alternative pathway of complement activation by acting as a cofactor for factor I (FI), facilitating the degradation of C3b and thereby controlling C3 convertase activity. [17,18] Given the ability of *K. kingae* strain KK01Δ*csaA*Δ*pam* to resist human serum killing via KK02920, we sought to determine whether FH bound to KK02920 was functionally active by examining C3b degradation. Strains KK01Δ*csaA*Δ*pam* and KK01Δ*csaA*Δ*pam*Δ*02920* at an inoculum of 1 x $10^6$ CFU were incubated with FH and then were incubated with C3b and FI and resolved by SDS-PAGE. As controls, we incubated bacteria in the absence of FH or FI. As demonstrated in Fig 8, the C3b degradation fragments α′ 43-kDa and α′ 41-kDa were absent in samples lacking FH or FI, indicating that both of these factors are necessary for C3b degradation. Incubation of strain KK01Δ*csaA*Δ*pam* with FH and FI produced the α′ 43-kDa and α′ 41-kDa C3b degradation fragments. In contrast, these degradation fragments were markedly reduced when strain KK01Δ*csaA*Δ*pam*Δ*02920* was incubated with FH and FI. These results suggest that FH retains cofactor activity when bound to *K. kingae* and is dependent on KK02920.

### A collection of *K. kingae* clinical isolates bind human FH to resist complement-mediated killing

To extend our results, we examined a sample of *K. kingae* clinical isolates for binding of FH. We selected a group of eight clinical isolates that were previously observed to exhibit moderate to high levels of serum resistance and that encompassed

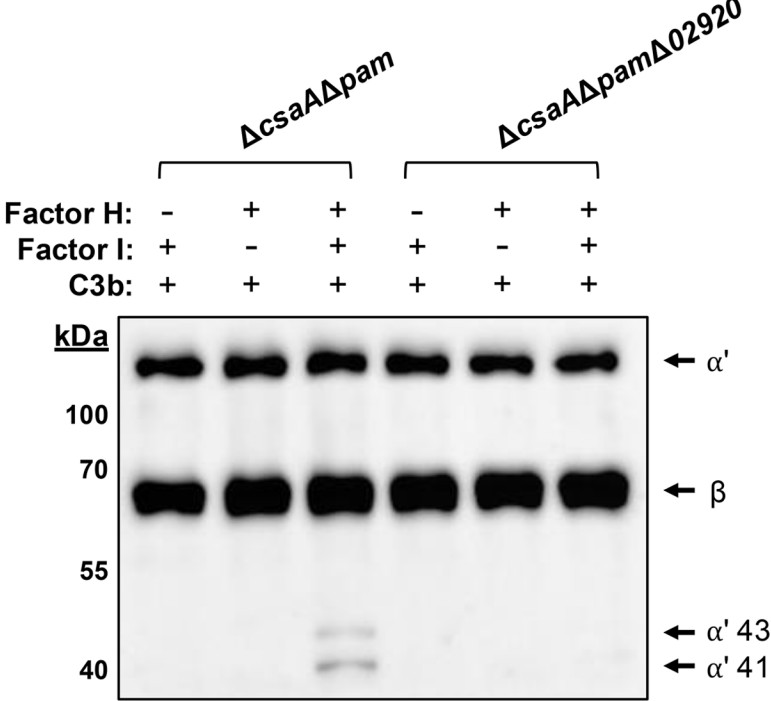

**Fig 8. FH retains cofactor activity when bound to *K. kingae*, and this activity is dependent on KK02920.** *K. kingae* strains KK01Δ*csaA*Δ*pam* and KK01Δ*csaA*Δ*pam*Δ*02920* at an inoculum of 1 x 10⁶ CFU were preincubated with 10 µg/mL FH for 1 h. Bacteria were subsequently incubated with 10 µg/mL C3b and 2 µg/mL factor I (FI) for 1 h. Samples without FH or FI were included as controls. Presence (+) and absence (-) of each component is listed above the image. Samples were separated on a 10% SDS-PAGE gel and transferred to a nitrocellulose membrane. A western blot was performed to detect the degradation of C3b by incubating the membrane with an anti-human C3 pAb and an HRP-coupled anti-goat IgG. The arrows indicate the position of the two intact C3b chains (α′ and β) and the degradation fragments (α′ 43-kDa and α′ 41-kDa). A representative image is shown.

carrier and invasive isolates across the *K. kingae* population. [8] Outer membrane protein preparations from the eight *K. kingae* clinical isolates were resolved by SDS-PAGE and subjected to far-western blot analysis with FH. As shown in Fig 9A, all eight isolates produced a band approximately the same molecular mass as KK02920 in strain KK01. Outer membrane protein preparations from isolates KK146 and PYKK58 produced comparatively weaker FH-binding bands.

To determine whether FH binding by the eight *K. kingae* clinical isolates contributes to serum resistance, we used rat serum as a source of active complement and incubated the isolates for 1 h with either 10% normal rat serum (NRS) or 10% heat-inactivated NRS (HI-NRS) as a source of inactive complement. Each condition was supplemented with either 0 µg/mL or 20 µg/mL purified human FH. Serum sensitivity was calculated by dividing the number of CFU recovered from NRS by the number of CFU recovered from HI-NRS. As shown in Fig 9B, all of the *K. kingae* clinical isolates incubated with 10% NRS and 0 µg/mL human FH exhibited virtually no survival, indicating that these isolates are susceptible to the bactericidal activity of rat complement. Supplementing rat serum with 20 µg/mL human FH restored survival for each of the *K. kingae* clinical isolates. These results suggest that KK02920 is widespread across the *K. kingae* population structure and can bind human FH to resist complement-mediated killing.

## Discussion

*Kingella kingae* is a common commensal organism in the oropharynx in young children and is the leading cause of bone and joint infections in this patient population. [1–3] To cause invasive disease, *K. kingae* must breach the respiratory epithelium, enter the bloodstream, resist intravascular innate immune mechanisms, and disseminate to distal sites. Previous

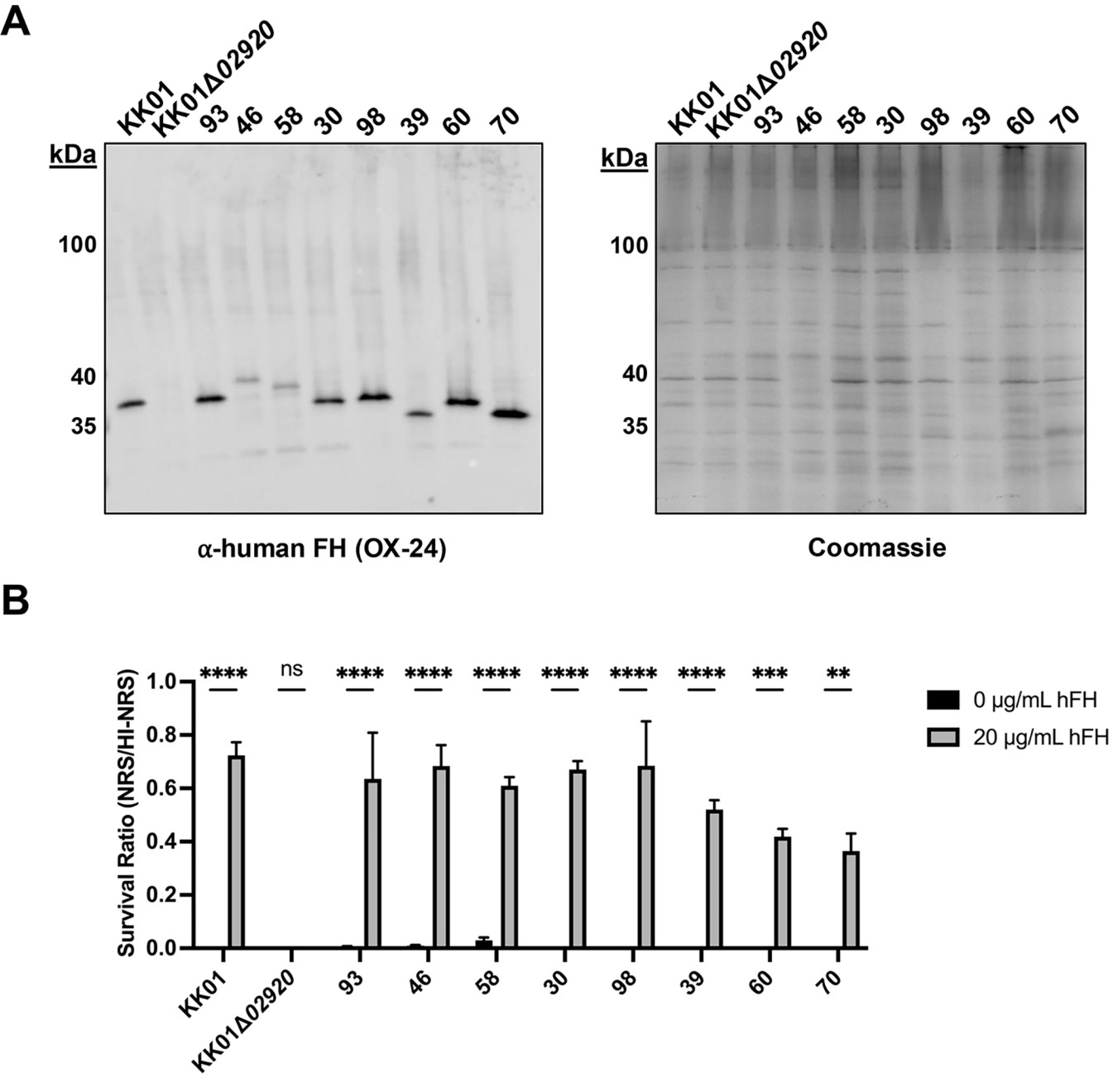

**Fig 9. A collection of _K. kingae_ clinical isolates bind human FH to resist rat complement-mediated killing.** _K. kingae_ strains KK01, KK01Δ_02920_, and clinical isolates were evaluated for (A) FH binding via far-western blot analysis and for (B) resistance to rat complement-mediated killing via FH binding. Abbreviations: 93, _K. kingae_ clinical isolate PYKK93; 46, _K. kingae_ clinical isolate KK146; 58, _K. kingae_ clinical isolate PYKK58; 30, _K. kingae_ clinical isolate ATCC 23330; 98, _K. kingae_ clinical isolate PYKK98; 39, _K. kingae_ clinical isolate E3339; 60, _K. kingae_ clinical isolate PYKK60. 70, _K. kingae_ clinical isolate BB270. (A) Outer membrane preparations were isolated and separated on a 12.5% SDS-PAGE gel and transferred to a nitrocellulose membrane or stained with Coomassie blue. A far-western blot was performed by incubating the membrane in 10% HI-NHS, anti-human FH mAb (OX-24), and an HRP-coupled anti-mouse IgG. The band between 35 and 40 kDa for each strain represents the outer membrane protein that binds human FH. The ~100-kDa band represents a loading control. A representative image is shown. (B) _K. kingae_ strains (~$10^3$ CFU) were incubated with 10% normal rat serum (NRS) or heat-inactivated NRS (HI-NRS) with either 0 µg/mL or 20 µg/mL of human factor H (hFH) for 1 h. The survival ratio was calculated by dividing NRS CFU counts by the HI-NRS CFU counts. A total of 3 biological replicates were performed (_n_ = 3). The survival ratios are negligible for bacteria incubated with 0 µg/mL hFH and for strain KKΔ_02920_ incubated with 20 µg/mL hFH. Data are presented as means, and the error bars represent the standard error of the mean. Statistical significance was determined by 2-way analysis of variance (ANOVA) with Sidak's correction for multiple comparisons. **, _P_ < 0.005; ***, _P_ < 0.0005; ****, _P_ < 0.0001.

work has established that *K. kingae* produces a polysaccharide capsule and an exopolysaccharide that aid in evading innate immune mechanisms, including serum-mediated killing, neutrophil-mediated killing, and antimicrobial peptide-mediated killing. [8,15]

In encapsulated organisms, the polysaccharide capsule is a major mechanism of serum resistance. Examples include *Neisseria meningitidis*, *Haemophilus influenzae* type b, and *Klebsiella pneumoniae,* where elimination of the capsule results in a marked decrease in serum resistance [35–39] and overexpression of capsule in these organisms results in increased serum resistance. [40–44] In some encapsulated bacteria, complement-binding proteins also contribute to serum resistance of encapsulated organisms. [45] In contrast, in nonencapsulated organisms such as *N. gonorrhoeae* and nontypeable *H. influenzae,* complement-binding proteins are the predominant drivers of serum resistance. [46–51] In our studies with *K. kingae*, elimination of both the capsule and the exopolysaccharide resulted in a very modest decrease in serum resistance, while elimination of the KK02920 human FH binding protein virtually abolished serum resistance, demonstrating that KK02920 is the predominant determinant of serum resistance.

Pathogenic organisms that bind FH often have multiple mechanisms of binding this negative complement regulator. [34] For example, *N. meningitidis* possesses 5 known mechanisms of FH binding, [52–56] *Streptococcus pneumoniae* possesses 4 known mechanisms of FH binding, [57–61] and *Candida albicans* possesses 4 known mechanisms of FH binding. [62–66] On the other hand, *H. influenzae* type b and nontypeable *H. influenzae* appear to possess only one known mechanism of FH binding. [67–69] In our study of *K. kingae*, mass spectrometry analysis of the 37-kDa band detected by far-western analysis of outer membrane proteins overlaid with FH identified two potential factor H binding proteins, namely KK02010 and KK02920. Primary amino acid sequence predictions for both KK02010 and KK02920 revealed predicted human factor H binding domains, but flow cytometry and far-western analyses with strains lacking one or the other of these proteins established that only KK02920 has FH binding activity. Further work is necessary to elucidate the function of KK02010 and the role of the predicted KK02010 factor H binding domain.

We observed that the ability of *K. kingae* to bind FH and resist complement-mediated killing was virtually abrogated by eliminating KK02920. Although it is tempting to attribute the entire serum resistance phenotype to KK02920 binding of FH, it is important to consider the possibility that the KK02920 protein also hijacks other complement regulators that contribute to overall serum resistance. Along these lines, there are several examples where one protein can bind multiple complement regulators. In particular, the *N. gonorrhoeae* Por1B protein [70,71] and the nontypeable *H. influenzae* P5 protein bind FH and C4b-binding protein (C4BP), [49,50,67] the *S. pneumoniae* Hic protein binds FH and vitronectin, [61,72] and the *S. pneumoniae* PspC protein binds FH, vitronectin, and C4BP. [57,58,73,74] In future studies we will examine whether KK02920 can bind other complement regulators beyond FH.

It is interesting that *K. kingae* possesses multiple factors that mediate resistance to complement-mediated serum killing. The results in this study in combination with our previous work [8] indicate that the capsule and exopolysaccharide inhibit serum killing primarily through the classical complement pathway, with a modest contribution to protection against the alternative pathway presumably related to reduced deposition of C3b. [8,11] Furthermore, our current findings suggest that KK02920 is critical for inhibiting serum killing via the alternative complement pathway by binding FH, which acts as a cofactor for factor I to promote the degradation of C3b on the bacterial surface. This redundancy suggests that resistance to complement-mediated killing may be important for *K. kingae* beyond conferring survival in the bloodstream, perhaps also playing a role in promoting survival in the oropharynx, the natural habitat of *K. kingae*. Indeed, it is unlikely that the evolution of *K. kingae* resistance to complement-mediated killing was driven by the need to survive in the intravascular space, a dead end for *K. kingae*. Instead, oropharyngeal secretions likely contain sufficient levels of complement during health and disease to threaten *K. kingae* viability. [6,7,75–77]

KK02920 is most closely related to factor H-binding protein (FHbp) of *Neisseria* spp., including commensal and pathogenic *Neisseria* species. [78–81] FHbp was initially identified as a potential vaccine antigen in *N. meningitidis,* reflecting its conservation among diverse *N. meningitidis* strains and its ability to stimulate bactericidal antibodies against

*N. meningitidis*. [78,80] This protein was subsequently found to bind FH and mediate resistance to complement-mediated serum killing and has been incorporated into two recently licensed protein-based vaccines against meningococcal serogroup B strains, Bexsero and Trumenba. [79,82–85] Given the critical role that KK02920 plays in promoting *K. kingae* survival in serum, this protein is an attractive potential vaccine antigen against *K. kingae* disease.

We demonstrated that representatives across the *K. kingae* population structure possess a FH binding band similar in molecular mass to KK02920, suggesting that KK02920 is broadly conserved across the *K. kingae* population structure. However, slight variations in band size and band intensity were observed. We speculate that amino acid sequence heterogeneity in KK02920 may influence the affinity for FH. Alternatively, there may be strain-to-strain differences in expression levels of KK02920. In support of these hypotheses, multiple allelic variants of the *N. meningitidis* FHbp have been identified and categorized into two distinct subfamilies designated subfamily A and subfamily B. [86] FHbp expression levels among meningococcal strains can vary at least 15-fold. [87] Although far-western blot analysis revealed variation in the FH binding profiles among these *K. kingae* clinical isolates, these differences did not appear to correlate with the ability of these isolates to utilize human FH to evade rat serum-mediated killing or to evade human serum-mediated killing. [8]

*K. kingae* initiates infection by colonizing the oropharynx. The first step in colonization and subsequent infection is adherence to host cells. Because complement regulators naturally bind to host cells, these proteins can serve as a bridge between bacteria and host cells, facilitating bacterial adherence. As examples, binding of FH to *S. pneumoniae* PspC and to *N. gonorrhoeae* Por1A increases bacterial adherence to host cells. [88–90] Additionally, the N terminus of vitronectin interacts with integrins and host cells while the C terminus of vitronectin interacts with bacteria, aiding in the adherence and internalization of bacteria such as *H. influenzae*, *N. gonorrhoeae*, *S. pneumoniae*, and *P. aeruginosa*. [91–97] Our lab has established that *K. kingae* adheres to human epithelial cells via type IV pili and a trimeric autotransporter called Knh. [98,99] These studies have involved an in vitro model that assesses adherence in the absence of FH and other human serum components, precluding recognition of a role for FH in *K. kingae* adherence.

In conclusion, our data demonstrate that *K. kingae* produces a factor H binding protein that is the predominant determinant of serum resistance and is important for invasive disease, revealing the importance of a complement regulator-binding protein for serum resistance in an encapsulated pathogen.

## Materials & methods

### Ethics statement

All animal experiments described within were conducted in accordance with the Animal Welfare Act, the Public Health Service policy on the humane care and use of laboratory animals from the U.S. Department of Health and Human Services, and the recommendations in the *Guide for the Care and Use of Laboratory Animals* of the National Research Council. [100] The Children's Hospital of Philadelphia animal research facilities have full accreditation from the Association for Assessment and Accreditation of Laboratory Animal Care (AAALAC) International. All animal procedures were approved by the Children's Hospital of Philadelphia Institutional Animal Care and Use Committee (IACUC) under protocol IAC 22–001050.

### Bacterial strains and growth conditions

The strains used in this study are listed in Table 1. *K. kingae* strains were stored at -80°C in brain heart infusion (BHI) broth with 20% glycerol. *E. coli* strains were stored at -80°C in Luria-Bertani (LB) broth with 15% glycerol. *K. kingae* strains were grown at 37°C with 5% $CO_2$ on chocolate agar, supplemented with 2 µg/mL tetracycline or 50 µg/mL kanamycin as appropriate. *E. coli* strains were grown at 37°C with 5% $CO_2$ on LB agar or shaking at 250 rpm in LB broth, supplemented with 100 µg/mL ampicillin as appropriate.

**Table 1. Strains and plasmids used in this study.**

| Strain or plasmid | Description | Reference or Source |
|---|---|---|
| *E. coli* strains | | |
| DH5α | *E. coli* F⁻ φ80d*lacZΔM15* Δ(*lacZYA-argF*)*U169 deoR recA1 endA1 hsdR17*(r_K⁻ m_K⁺) *phoA supE441 thi-1 gyrA96 relA1* | [101] |
| XL-10 Gold | TetʳΔ(*mcrA*)*183* Δ(*mcrCB-hsdSMR-mrr*)*173 endA1 supE44 thi-1 recA1 gyrA96 relA1 lac* Hte [F´ *proAB lacIqZΔM15* Tn*10* (Tetr) Amy Camʳ] | Agilent |
| *K. kingae* strains | | |
| KK01 | Naturally occurring nonspreading and noncorroding variant of septic arthritis clinical isolate 269–492 | [7] |
| KK01Δ*csaA* | Contains an unmarked *csaA* deletion | [26] |
| KK01Δ*pam* | Contains an *ermC* marked pamABCDE deletion | [27] |
| KK01Δ*csaA*Δ*pam* | Contains an unmarked *csaA* deletion and an *ermC* marked *pamABCDE* deletion | [26] |
| KK01Δ*02010* | Contains an *aphA3* marked *02010* disruption | This study |
| KK01Δ*02920* | Contains a *tetM* marked *02920* disruption | This study |
| KK01Δ*csaA*Δ*pam*Δ*02010* | Contains an unmarked *csaA* deletion, an *ermC* marked *pamABCDE* deletion, and an *aphA3* marked *02010* disruption | This study |
| KK01Δ*csaA*Δ*pam*Δ*02920* | Contains an unmarked *csaA* deletion, an *ermC* marked *pamABCDE* deletion, and a *tetM* marked *02920* disruption | This study |
| KK01Δ*02920*(*02920*) | Contains an *aphA3* marked complement of *02920* in the Δ*02920* background | This study |
| KK01Δ*csaA*Δ*pam*Δ*02920*(*02920*) | Contains an *aphA3* marked complement of *02920* in the Δ*csaA*Δ*pam*Δ*02920* background | This study |
| PYKK93 | Clonal group P isolate from a case of bacteremia | P. Yagupsky |
| KK146 | Clonal group N isolate from a case of bacteremia | P. Yagupsky |
| PYKK58 | Clonal group N isolate from a case of septic arthritis | P. Yagupsky |
| ATCC 23330 | Clonal group D isolate from a healthy carrier | ATCC |
| PYKK98 | Clonal group B isolate from a case of bacteremia | P. Yagupsky |
| E3339 | Clonal group F isolate from a healthy carrier | P. Yagupsky |
| PYKK60 | Clonal group D isolate from a case of endocarditis | P. Yagupsky |
| BB270 | Clonal group U isolate from a healthy carrier | P. Yagupsky |
| Plasmids | | |
| pFalcon2 | Source of *aphA3* kanamycin resistance marker | [102] |
| pHSX*tetM*4 | Source of *tetM* kanamycin resistance marker | [103] |
| pUC19/Δ*02010:aphA3* | *02010* disruption construct with an *aphA3* kanamycin resistance marker | This study |
| pUC19/Δ*02920:tetM* | *02920* disruption construct with a *tetM* tetracycline resistance marker | This study |
| pCKan/(*02920*) | *02920* disruption construct with a *aphA3* kanamycin resistance marker | This study |

### *K. kingae* mutant strain construction

*K. kingae* plasmid-based gene disruption and complementation constructs were generated as previously described. [98,104] Briefly, plasmid-based gene disruption constructs were created in *E. coli*, linearized, and introduced into *K. kingae* via natural transformation, followed by recovery of transformants by plating onto chocolate agar containing appropriate antibiotics. Correct gene disruptions and complementation constructs were confirmed by PCR amplification and Sanger sequencing of genomic DNA prepared from putative mutant strains. All primers mentioned are listed in Table 2.

To create the KK*02010* disruption construct, a fragment spanning approximately 1000-bp immediately upstream of the KK*02010* ORF was amplified from strain KK01 using the primers Δ*02010*_5′_F and Δ*02010*_5′_R. The resulting amplicon was digested with EcoRI/KpnI and ligated into EcoRI/KpnI-digested pUC19, generating pUC19/Δ*02010*_5′. A fragment spanning approximately 1000-bp immediately downstream of the KK*02010* ORF was amplified from strain KK01 using the primers Δ*02010*_3′_F and Δ*02010*_3′_R. The resulting amplicon was digested with BamHI/HindIII and ligated into BamHI/HindIII-digested pUC19/Δ*02010*_5′, generating pUC19/Δ*02010*_5′-3′. The kanamycin resistance cassette *aphA3* was amplified from pFalcon2 using the primers *aphA3*_F and *aphA3*_R. The kanamycin resistance cassette was digested with KpnI/BamHI and ligated into KpnI/BamHI-digested pUC19/Δ*02010*_5′-3′, generating pUC19/Δ*02010*:*aphA3*. To generate strains KK01Δ*02010* and KK01Δ*csaA*Δ*pam*Δ*02010*, the plasmid was linearized and transformed into strains KK01 and KK01Δ*csaA*Δ*pam* via natural transformation. Transformants were recovered on chocolate agar containing 50 μg/mL kanamycin.

To create the KK*02920* disruption construct, a fragment spanning approximately 250-bp immediately upstream of the KK*02920* ORF to approximately halfway through the ORF was amplified from strain KK01 using the primers Δ*02920*_up_F and Δ*02920*_up_R. A fragment spanning the second half of the ORF to approximately 250-bp immediately downstream

**Table 2. Primers used in this study.**

| Primer Name | Sequence (5′- 3′) |
| --- | --- |
| Δ*02010*_5′_F | ACGTGAATTCAAACCCGTGCGCGACAAATG |
| Δ*02010*_5′_R | ACGTGGTACCTCAAAAGTGCCAATAAAGTTTTG |
| Δ*02010*_3′_F | ACGTGGATCCGTGGTAAACGCTAATTTATA-ATCTTG |
| Δ*02010*_3′_R | ACGTAAGCTTGCACAGAGTTCTCATCGTAATC |
| *aphA3*_F | GCATGGATCCCATCTAAATCTAGGTACTAAAA-CAATTCATCCAG |
| *aphA3*_R | GCATGGTACCGTTTGACAGCTTATCATCGATA-AACCCAG |
| Δ*02920*_up_F | AACAGCTATGACCATGATTACGCCATTT-GAAATTTTGTAAAACATTTCC |
| Δ*02920*_up_R | ATTGACAGTTTATTGGTTATATACTTTATAAGT-GCTATC |
| Δ*02920*_down_F | TATATAAATATGGGCCGTTTTCGTTTGTGGC |
| Δ*02920*_down_R | CGACGTTGTAAAACGACGGCCAGTGCTGAAT-CACATTGCATTAAGGC |
| Δ*02920*_tetM_F | AGTATATAACCAATAAACTGTCAATTTGATAGCG |
| Δ*02920*_tetM_R | AACGAAAACGGCCCATATTTATATAACAACATA-AAATACAC |
| *02920*_comp_F | GATAAGCTGTCAAACGGTACAGCGGCTGAT-GAACATAG |
| *02920*_comp_R | CAAGCTTGCATGCCTGCAGGTTCTGTTGCATT-AGGACC |

of the ORF was amplified from strain KK01 using the primers Δ*02920*_down_F and Δ*02920*_down_R. The tetracycline resistance cassette *tetM* was amplified from the plasmid pHSX-*tetM*4 using the primers Δ*02920_tetM*_F and Δ*02920_tetM*_R. The fragments were assembled in EcoRI/HindIII-digested pUC19 using NEBuilder Hi-Fi DNA Assembly Mastermix (New England Biolabs, Ipswich, MA) to generate pUC19/Δ*02920*. To generate strains KK01Δ*02920* and KK01Δ*csaA*Δ*pam*Δ*02920*, the plasmid was linearized with SfoI and transformed into strains KK01 and KK01Δ*csaA*Δ*pam* via natural transformation. Transformants were recovered on chocolate agar containing 2 µg/mL tetracycline.

To complement the KK*02920* disruption, a fragment containing the KK*02920* ORF and approximately 300-bp upstream to include the promoter was PCR amplified using the primers *02920*_comp_F and *02920*_comp_R. The resulting fragment was ligated into SalI/KpnI-digested pCKan, which is a modified version of the previously described *K. kingae* complementation construct pCErm [98] that contains a kanamycin resistance cassette, to generate pCKan/(*02920*). To generate strain KK01Δ*02920*(02920) and KK01Δ*csaA*Δ*pam*Δ*02920*(02920), pCKan/(*02920*) was digested with SfoI and introduced into strains KK01Δ*02920* and KK01Δ*csaA*Δ*pam*Δ*02920*, respectively, and the transformants were recovered on chocolate agar containing 50 µg/mL kanamycin.

## Flow cytometry analysis

Binding of FH via flow cytometry was determined as previously described. [8] In brief, *K. kingae* and *E. coli* strains were grown on chocolate agar and LB agar plates, respectively, and resuspended in PBS. Bacterial samples of 2 x 10^8 colony forming units (CFU) were pelleted, washed once in PBS, resuspended in PBS, and incubated with varying concentrations of heat inactivated normal human serum (HI-NHS) (pooled human complement serum, Innovative Research, Novi, MI) as a source of FH for 1 h at room temperature (RT) with gentle agitation. Bacteria were washed once with PBS and resuspended in 4% paraformaldehyde in PBS for fixation. After incubation for 30 minutes at RT, bacteria were washed twice with Tris-buffered saline (TBS) and resuspended in TBS containing 50 mM EDTA and 0.1% bovine serum albumin (BSA). Fixed bacteria were incubated with a 1:250 dilution of a mouse anti-human FH monoclonal antibody OX-24 (Invitrogen Thermo Fisher, Rockford, IL) for 1h rotating at RT. Samples were washed twice with PBS, resuspended in PBS containing 0.1% BSA, and incubated with a 1:200 dilution of a polyclonal rabbit anti-mouse IgG DyLight 488-conjugated antibody (Rockland, Limerick, PA) for 45 min rotating at RT. Bacteria were washed twice with PBS, resuspended in PBS, and stained with propidium iodide (PI) (Biotium, Fremont, CA) for flow cytometry analysis. Flow cytometry assays were performed using an CytoFLEX S (3 lasers) instrument (Beckman Coulter, Brea, CA).

## Serum bactericidal assays

Serum bactericidal assays were conducted as previously described with minor modifications. [8] In brief, *K. kingae* strains and *E. coli* strain DH5α were grown on chocolate agar and LB agar plates, respectively, and then resuspended in PBS containing 0.1% gelatin (PBS-G). For assays using human serum, samples were diluted in PBS-G to obtain a concentration of approximately 1.0 x 10^8 CFU/mL, and 10 µL were used for a final inoculum of 1.0 x 10^6 CFU. The inocula were mixed with normal human serum (NHS) (pooled human complement serum, Innovative Research) or heat-inactivated NHS (HI-NHS) (prepared by incubating NHS at 56°C for 30 min) to a final serum concentration of 50%. For assays using rat serum, samples were diluted in PBS-G to obtain a concentration of approximately 1.0 x 10^5 CFU/mL, and 10 µL were used for a final inoculum of 1.0 x 10^3 CFU. The inocula were mixed with various concentrations of normal rat serum (NRS) (Cocalico Biologics Inc., Stevens PA) or heat-inactivated NRS (HI-NRS), mixed with various concentrations of purified human FH (Complement Technology, Tyler, TX) diluted in PBS-G, as appropriate. Samples were incubated for 1 h at 37°C with 5% $CO_2$. Serial dilutions of the inoculum and reaction samples were plated on chocolate agar and incubated overnight at 37°C with 5% $CO_2$ to determine the CFU counts. To perform classical pathway inhibition assays, samples were incubated with EGTA at a final concentration of 5 mM and supplemental $MgCl_2$ at a final concentration of 9 mM prior to introduction of NHS or NRS.

## Juvenile rat infection model

Experiments using the juvenile rat infection model of *K. kingae* disease were conducted as previously described. [8] *K. kingae* strains were grown on chocolate agar plates for ~18 h, and the bacteria were swabbed from plates and resuspended in PBS to a final density of $1 \times 10^8$ CFU/mL. Bacterial suspensions of 100 µL ($1 \times 10^7$ CFU) were supplemented with purified human FH to achieve a final concentration of 50 µg purified human FH (Complement Technology) per 150 µL aliquot (333 µg/mL), or supplemented with PBS as a control. Nursing 5-day-old Sprague Dawley rat pups (Charles River Laboratories, Wilmington, MA) were injected via the intraperitoneal route with 150 µL containing bacteria alone or bacteria co-incubated with 50 µg purified human FH. As controls, rat pups were injected with PBS alone or 50 µg purified human FH. Rat pups were injected using a 27-gauge needle and then returned to their cage with a lactating dam. The experimental and control groups were housed separately with a lactating dam and were monitored for mortality and signs of illness every 6 h for the first 30 h and then twice daily for a total of 5 days. Animals found to be moribund were euthanized by using $CO_2$ inhalation followed by secondary decapitation.

## Far-western blot analysis

Outer membrane fractions were isolated from whole-cell bacterial sonicates on the basis of sarkosyl insolubility as described by Carlone et al. [105] Samples were standardized by protein concentration, resolved by 12.5% SDS-PAGE, and then either visualized by Coomassie blue staining or transferred to nitrocellulose membranes. Membranes were blocked with 5% milk in PBS (blocking buffer) overnight at 4°C and were probed with FH, either 100 µg/mL purified human FH (Complement Technology) or 10% HI-NHS (pooled human complement serum, Innovative Research), in blocking buffer for 2 h shaking at RT. Following washing with Tris-buffered saline with 0.1% Tween-20 (TBST) three times for 5 min, the membranes were incubated with 1:1000 diluted mouse anti-human FH monoclonal antibody OX-24 (Invitrogen Thermo Fisher) in blocking buffer overnight at 4°C with gentle agitation. After washing with TBST, the membranes were incubated with 1:5000 diluted anti-mouse IgG horseradish peroxidase (HRP)-conjugated antibody in blocking buffer for 1 h shaking at RT, and subsequently washed with TBST. Far-western blot membranes were developed with SuperSignal West Pico PLUS chemiluminescent blotting substrate (ThermoScientific, Rockford, IL) and imaged using a G-box Chemi:XX6 system (Syngene, Frederick, MD).

## Mass spectrometry

**In-gel digestion.** The sample was run into an SDS-PAGE gel and stained using Coomassie G250. The band of interest was excised from the gel and cut into $1 mm^3$ cubes. The gel pieces were destained with 50% methanol/2.5% acetic acid, reduced with 5mM dithiothreitol (Thermo), and alkylated with 20mM iodoacetamide (Sigma). After alkylation, gel pieces were washed with 50 mM ammonium bicarbonate (Sigma) and dehydrated with acetonitrile (Fisher). For enzymatic hydrolysis, 5 ng/µL trypsin (Promega) in 50mM ammonium bicarbonate/20% acetontirile was added to the gel pieces and incubated overnight at 37°C. Peptides were extracted with 0.3% trifluoroacetic acid (TFA) (Pierce) followed by 50% acetonitrile. The extracts were combined and volume reduced to remove acetonitrile by vacuum centrifugation. Peptides were desalted using a C18 stagetip, then dried by vacuum centrifugation and reconstituted in 0.1% TFA containing iRT peptides (Biognosys, Schlieren, Switzerland).

**Mass spectrometry data acquisition.** Sample was analyzed on a Q-Exactive HF mass spectrometer (Thermofisher Scientific San Jose, CA) coupled with an Ultimate 3000 nano UPLC system and an EasySpray source. 5 µL of sample was loaded onto an Acclaim PepMap 100 75 µm x 2 cm trap column (Thermo) at 5 µL/min, and separated by reverse phase (RP)-HPLC on a nanocapillary column, 75 µm id × 50 cm 2 µm PepMap RSLC C18 column (Thermo). Mobile phase A consisted of 0.1% formic acid and mobile phase B of 0.1% formic acid/acetonitrile. Peptides were eluted into the mass spectrometer at 300 nL/min with each RP-LC run comprising a 90-minute gradient from 3% B to 45% B.

The mass spectrometer was set to repetitively scan m/z from 300 to 1400 (R = 240,000) followed by data-dependent MS/MS scans on the twenty most abundant ions with minimum automatic gain control (AGC) 1e4, dynamic exclusion with a repeat count of 1, repeat duration of 30s, and resolution of 15000. The AGC target value was 3e6 and 1e5, for full and MSn scans, respectively. MSn injection time was 160 ms. Rejection of unassigned and 1 +, 7–8 + charge states was set.

**Mass spectrometry QA/QC and system suitability.** The suitability of the instrumentation was monitored using QuiC software (Biognosys; Schlieren, Switzerland) for the analysis of the spiked-in iRT peptides. As a measure for quality control, we injected standard E. coli protein digest before and after the sample set and collected the data in data dependent acquisition (DDA) mode. The collected DDA data were analyzed in MaxQuant [106] and the output was subsequently visualized using the PTXQC [107] package to track the quality of the instrumentation.

**Database searching.** All MS/MS samples were analyzed using MSFragger (The Nesvizhskii Lab, 1301 Catherine, 4237 Medical Science I, Ann Arbor, MI 48109; version 3.8). MSFragger was set up to search a reverse concatenated *Kingella kingae* strain KK01 protein sequence database (1962 entries) assuming the digestion enzyme trypsin. MSFragger was searched with a fragment ion mass tolerance of 20 PPM and a parent ion tolerance of 20 PPM. Carbamidomethyl of cysteine was specified in MSFragger as a fixed modification. Oxidation of methionine and acetyl of the n-terminus were specified as variable modifications.

Scaffold (version Scaffold_5.3.3, Proteome Software Inc., Portland, OR) was used to validate MS/MS based peptide and protein identifications. Peptide identifications were accepted if they could be established at greater than 98.0% probability to achieve an FDR less than 0.1% by the Percolator posterior error probability calculation. [108] Protein identifications were accepted if they could be established at greater than 99.0% probability to achieve an FDR less than 1.0% and contained at least 2 identified peptides. Protein probabilities were assigned by the Protein Prophet algorithm. [109] Proteins that contained similar peptides and could not be differentiated based on MS/MS analysis alone were grouped to satisfy the principles of parsimony. Proteins sharing significant peptide evidence were grouped into clusters.

### Capsule extraction and visualization

Capsule was extracted as previously described with minor modifications. [98,110] Briefly, bacteria were resuspended in 3 mL PBS to an $OD_{600}$ of 1.0, centrifuged for 5 min at 20,000 x g, and resuspended in 1 mL 50mM Tris acetate (pH 5). After agitation for 30 min while rotating, bacteria were removed by centrifugation. The capsule extracts were concentrated and treated with proteinase K overnight at 37°C. Capsule extracts were resolved by 7.5% SDS-PAGE and visualized with Alcian blue staining.

### Galactan exopolysaccharide extraction and visualization

Bacteria were cultured for approximately 20 h on chocolate agar and suspended in 3 mL PBS to an $OD_{600}$ of 1.0. Following gentle agitation for 30 min at ambient temperature, the bacteria were removed by centrifugation (2 min at 8000 × g), and 1 mL of the supernatant was concentrated to approximately 30 μL over a 100,000-molecular weight cutoff (MWCO) Amicon Ultra centrifugal filter (MilliporeSigma, Burlington, MA, USA). The samples were treated with 20 μg of proteinase K for 1 h at 55°C and were then separated using 16.5% DOC-PAGE gel electrophoresis and probed with an anti-galactan antiserum by Western analysis as previously described. [26]

### Cofactor assay

Cofactor activity of *K. kingae* was performed as previously described for *H. influenzae*. [69,111]. In brief, *K. kingae* strains were grown on chocolate agar and resuspended in PBS. Bacterial samples of 1 x $10^6$ CFU were pelleted, washed once in PBS, resuspended in PBS, and incubated with 10 μg/mL purified human FH for 1 h at 37°C with gentle agitation. Bacteria were washed 4 times with PBS, resuspended in PBS, and incubated with 10 μg/mL purified human C3b (Complement Technology) and 2 μg/mL purified human factor I (Complement Technology) for 1 h at 37°C with gentle agitation. The reactions

were terminated by the addition of SDS-PAGE sample buffer. The samples were resolved by 10% SDS-PAGE, transferred to nitrocellulose membranes, and blocked with 5% milk in PBS (blocking buffer) for 1h shaking at RT. The membranes were incubated with 1:1000 diluted goat anti-human C3 polyclonal antibody (Complement Technology) in blocking buffer overnight at 4°C with gentle agitation. After washing with Tris-buffered saline with TBST three times for 5min, the membranes were incubated with 1:5000 diluted anti-goat IgG horseradish peroxidase (HRP)-conjugated antibody in blocking buffer for 1h shaking at RT and were subsequently washed with TBST. Western blot membranes were developed with SuperSignal West Pico PLUS chemiluminescent blotting substrate (ThermoScientific) and imaged using a G-box Chemi:XX6 system (Syngene).

## Statistical analysis

Statistical analyses were performed with GraphPad Prism software for Mac (version 10.4.1; GraphPad Software, San Diego, CA). A $P$ value of <0.05 was considered statistically significant. The specific statistical tests used for each experiment are specified in the relevant figure legend.

## Supporting information

**S1 Fig: Supplementing human FH does not affect bactericidal activity of rat complement.** *K. kingae* strain KK01 and *E. coli* strain DH5α [~$10^3$ colony forming units (CFU)] were incubated with 5% normal rat serum (NRS) or heat-inactivated NRS (HI-NRS) with 0, 0.25, 1, or 100 μg/mL human FH. The survival ratio was calculated by dividing NRS CFU counts by the HI-NRS CFU counts. A total of 3 biological replicates were performed ($n=3$). Data are presented as means, and the error bars represent the standard error of the mean.
(TIFF)

**S2 Fig. *K. kingae* requires minimal concentrations of human FH to resist rat complement-mediated killing.** *K. kingae* strain KK01 (~$10^3$ CFU) was incubated with 5% NRS or HI-NRS with increasing concentrations of human FH, 0 μg/mL – 100 μg/mL. The survival ratio was calculated by dividing NRS CFU counts by the HI-NRS CFU counts. A total of 3 biological replicates were performed ($n=3$). Data are presented as means, and the error bars represent the standard error of the mean.
(TIFF)

**S3 Fig: Inhibition of the classical and lectin pathways restores *K. kingae* survival in rat serum.** *K. kingae* strain KK01 ($10^3$ CFU) was incubated with either 5% NRS or 5% HI-NRS alone, 5% NRS or 5% HI-NRS plus 9mM $Mg^{2+}$, 5% NRS or 5% HI-NRS plus EGTA, or 5% NRS or 5% HI-NRS plus EGTA and 9mM $Mg^{2+}$. The survival ratio was calculated by dividing NRS CFU counts by the HI-NRS CFU counts. A total of 3 biological replicates were performed ($n=3$). Data are presented as means, and the error bars represent the standard error of the mean. Statistical significance was determined by 1-way analysis of variance (ANOVA) with Tukey's correction for multiple comparisons. ***, $P<0.001$.
(TIFF)

**S4 Fig. Deletion of KK02920 does not affect *K. kingae* capsule or exopolysaccharide production.** (A) Capsular material was extracted from *K. kingae* strains KK01 and KK01Δ*02920*, separated by 7.5% SDS-PAGE and then stained with the cationic dye alcian blue. *K. kingae* strains KK01Δ*csaA*Δ*pam* and KK01Δ*csaA*Δ*pam*Δ*02920* were included as controls for no production of capsular material. The high molecular mass alcian blue-reactive material is indicative of the capsular material. A representative image is shown. (B) Galactan exopolysaccharide material was extracted from *K. kingae* strains KK01 and KK01Δ*02920*, separated by 16.5% DOC-PAGE, and transferred to a nitrocellulose membrane. A Western blot was performed by incubating the membrane with a galactan exopolysaccharide antiserum (GP-19) and anti-guinea pig HRP. The reactive mass spanning between 55 and 170kDa is representative of the galactan exopolysaccharide. A representative image is shown.
(TIF)

## Acknowledgments

We thank the Children's Hospital of Philadelphia Research Institute-UPENN Proteomics Core Facility (RRID:SCR_023099) for performing the mass spectrometry analyses. We thank Pablo Yagupsky at the Soroka University Medical Center for providing us with the *K. kingae* clinical isolates used in this study.

## Author contributions

**Conceptualization:** Kevin A. Hernandez, Eric A. Porsch, Joseph W. St. Geme III.

**Data curation:** Kevin A. Hernandez, Eric A. Porsch.

**Formal analysis:** Kevin A. Hernandez, Eric A. Porsch, Joseph W. St. Geme III.

**Funding acquisition:** Kevin A. Hernandez, Vanessa L. Muñoz, Joseph W. St. Geme III.

**Investigation:** Kevin A. Hernandez, Eric A. Porsch, Vanessa L. Muñoz.

**Methodology:** Kevin A. Hernandez, Eric A. Porsch, Vanessa L. Muñoz, Joseph W. St. Geme III.

**Project administration:** Eric A. Porsch, Joseph W. St. Geme III.

**Resources:** Joseph W. St. Geme III.

**Supervision:** Eric A. Porsch, Joseph W. St. Geme III.

**Validation:** Kevin A. Hernandez, Eric A. Porsch, Joseph W. St. Geme III.

**Visualization:** Kevin A. Hernandez.

**Writing – original draft:** Kevin A. Hernandez, Joseph W. St. Geme III.

**Writing – review & editing:** Kevin A. Hernandez, Eric A. Porsch, Vanessa L. Muñoz, Joseph W. St. Geme III.

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
