## [Decision Letter · Decision Letter 0]

30 Apr 2025

Identification of a Kingella kingae factor H binding protein that is the major determinant of serum resistance

PLOS Pathogens

Dear Dr. St. Geme III,

Thank you for submitting your manuscript to PLOS Pathogens. After careful consideration, we feel that it has merit but does not fully meet PLOS Pathogens's publication criteria as it currently stands. Therefore, we invite you to submit a revised version of the manuscript that addresses the points raised during the review process.

Please submit your revised manuscript within 60 days Jun 29 2025 11:59PM. If you will need more time than this to complete your revisions, please reply to this message or contact the journal office at plospathogens@plos.org. Please include the following items when submitting your revised manuscript:

We look forward to receiving your revised manuscript.

Kind regards,

Mathieu Coureuil

Guest Editor

PLOS Pathogens

Matthew Wolfgang

Section Editor

Editor-in-Chief

PLOS Pathogens

orcid.org/0000-0003-2946-9497

Editor-in-Chief

PLOS Pathogens

orcid.org/0000-0002-7699-2064

**Journal Requirements:**

At this stage, the following Authors/Authors require contributions: Kevin A. Hernandez, Eric A. Porsch, Vanessa L. Munoz, and Joseph W. St. Geme III. Please ensure that the full contributions of each author are acknowledged in the "Add/Edit/Remove Authors" section of our submission form.

**Reviewers' Comments:**

Reviewer's Responses to Questions

**Part I - Summary**

Reviewer #1: In previous research studies, St Geme et al. have demonstrated that the polysaccharide capsule and exopolysaccharide of Kingella kingae inhibit opsonin deposition and mediate resistance to the complement-mediated killing of the bacterium by human serum. However, mutants lacking capsule and exopolysaccharide still maintain some degree of complement resistance, suggesting that additional complement-resistant mechanism (s) exist. Using a series of elegant and precise experiments, the same research group identified the binding of the human serum factor H, a negative regulator of the alternative complement pathway, as responsible for the residual complement resistance activity once the K. kingae capsule and exopolysaccharide have been removed. Multiple virulence factors implicated in the resistance to complement-mediated bacterial killing emphasize the importance of complement resistance in the pathogenesis of oropharyngeal colonization and survival of the pathogen in the inhospitable bloodstream environment and target human tissues. In addition, the authors identified an outer-membrane protein named KK02920 as the bacterial component responsible for the fH binding. A similar protein (fHbp) is also responsible for binding fH in Neisseria meningitidis and has been incorporated into two recently licensed meningococcus serogroup B vaccines. In summary, the study not only adds to our current understanding of the mechanisms employed by K. kingae to subvert the host's immune response but also has identified a novel potential target for a future protective pediatric vaccine.

Major comment

The potential value of KK02920 as a component of a protective pediatric vaccine raises a few questions: Kingella kingae exhibits a wide range of genomic diversity, and the organism population comprises strains with remarkable differences in virulence. While some isolates are frequently carried on the oropharynx but are seldom if ever, recovered from invasive infections, a few others are responsible for the vast majority of osteoarthritis, bacteremia, and endocarditis cases. Following this line of thought: 1) what was the clinical source of the original (wild-type) KK01 strain (asymptomatic carriage?/invasive disease?). 2) It will be important to examine other K. kingae strains, invasive and mere respiratory colonizers, for the presence of KK02920. The potential significance of KK02920 as a target for such a vaccine will depend on its prevalence in the K. kingae population and, particularly, among strains of well-recognized invasiveness.

Reviewer #2: This paper shows that K. kingae can bind human FH through a protein called KK02920. Human FH can restore virulence of K. kingae in infant rats. The experiments are rigorous and the results are presented clearly.

Reviewer #3: In the study presented by Hernandez et al under the title “Identification of a Kingella kingae factor H binding protein that is the major determinant of serum resistance”, the authors present a good set of experiments that demonstrate that K. kingae binds the alternative complement pathway regulator factor H and that this binding could be responsible for protecting the microbe from complement-mediated killing. The design of the experiments is logical, and the interpretation of the results is mostly justified. In addition, the writing and the flow of the text are acceptable. However, I have some concerns regarding both the methodology and the conclusions, and addressing them could potentially improve the study substantially.

**Part II – Major Issues: Key Experiments Required for Acceptance**

Reviewer #1: In previous research studies, St Geme et al. have demonstrated that the polysaccharide capsule and exopolysaccharide of Kingella kingae inhibit opsonin deposition and mediate resistance to the complement-mediated killing of the bacterium by human serum. However, mutants lacking capsule and exopolysaccharide still maintain some degree of complement resistance, suggesting that additional complement-resistant mechanism (s) exist. Using a series of elegant and precise experiments, the same research group identified the binding of the human serum factor H, a negative regulator of the alternative complement pathway, as responsible for the residual complement resistance activity once the K. kingae capsule and exopolysaccharide have been removed. Multiple virulence factors implicated in the resistance to complement-mediated bacterial killing emphasize the importance of complement resistance in the pathogenesis of oropharyngeal colonization and survival of the pathogen in the inhospitable bloodstream environment and target human tissues. In addition, the authors identified an outer-membrane protein named KK02920 as the bacterial component responsible for the fH binding. A similar protein (fHbp) is also responsible for binding fH in Neisseria meningitidis and has been incorporated into two recently licensed meningococcus serogroup B vaccines. In summary, the study not only adds to our current understanding of the mechanisms employed by K. kingae to subvert the host's immune response but also has identified a novel potential target for a future protective pediatric vaccine.

Major comment

The potential value of KK02920 as a component of a protective pediatric vaccine raises a few questions: Kingella kingae exhibits a wide range of genomic diversity, and the organism population comprises strains with remarkable differences in virulence. While some isolates are frequently carried on the oropharynx but are seldom if ever, recovered from invasive infections, a few others are responsible for the vast majority of osteoarthritis, bacteremia, and endocarditis cases. Following this line of thought: 1) what was the clinical source of the original (wild-type) KK01 strain (asymptomatic carriage?/invasive disease?). 2) It will be important to examine other K. kingae strains, invasive and mere respiratory colonizers, for the presence of KK02920. The potential significance of KK02920 as a target for such a vaccine will depend on its prevalence in the K. kingae population and, particularly, among strains of well-recognized invasiveness.

Reviewer #2: None.

Reviewer #3: Major:

- The differentiation of which complement pathway is largely responsible for K. kingae killing was determined. Factor H is an alternative pathway regulator. The authors need to demonstrate that this killing is mediated via this pathway, not the other two. Standards assays are well-established in the field to make this differentiation. The previous study published by the group demonstrated that substantial killing is antibody-mediated (i.e. classical pathway).

- The authors demonstrated the binding of fH. Still, they did not demonstrate that this bound fH is functional (Cofactor Activity for Factor I cleaving C3b and Decay-Accelerating Activity displacing Bb from the C3 convertase). Standard assays are well-established in the field to prove this functionality.

- For the flow cytometry experiments, could the author provide representative histograms of the obtained results, especially for the experiments described in Figure 1?

- In Figure 2B, what is considered the 100% survival?

- What is the identified target protein conservation among different K. kingae strains?

- The binding and protection need to be tested among a diverse panel of K. kingae strains to demonstrate the generability of the observed phenotypes.

**Part III – Minor Issues: Editorial and Data Presentation Modifications**

Reviewer #1: (No Response)

Reviewer #2: This paper shows that K. kingae can bind human FH through a protein called KK02920. Human FH can restore virulence of K. kingae in infant rats. The experiments are rigorous and the results are presented clearly. Minor comments, mostly editorial, are presented below for the authors’ consideration.

1. Suggest using FH instead of fH to be consistent with the recommendations of the International Complement Society ( PMID: 31231398).

2. Line 17-20: May be worth also mentioning that the AP can also be triggered by C3b deposited through the CP and LP. In fact, spontaneous activation (alone) is very inefficient (PMID: 36424888).

3. The authors should provide representative flow cytometry tracings or data shown in Fig. 1.

4. If mAb OX-24 was used to detect FH binding when bacteria or western blot membranes were incubated with human serum (HI-NHS), please state either FH and/or the alternatively spliced molecule, FHL-1 could have been detected (PMID: 2936333). But Fig. 4 with pure FH suggests unequivocally that full length FH can bind to KK02920, as do the ‘rescue’ experiments with pure FH in juvenile rats.

5. What was the age of the rats from which rat serum was prepared (lines 371-372)? It is remarkable that even 0.25 mcg/mL of human FH can rescue the bacteria from killing by rat serum, which would not be expected if killing was driven by the classical pathway.

6. In light if the comment above it may be worth performing serum bactericidal assays in NHS and NRS with the csaA/pam KO strains where only the AP is active – for eg, using Mg-EGTA-treated serum.

Reviewer #3: Minor

Gram should be capitalized throughout the manuscript.

In vivo should not be italicized.

Western should not be capitalized.

The prime sign should be used instead of the apostrophe sign.

Bacterial species names should be italicized, for example, on line 449.

Please be consistent; sometimes, a space is left between the numerical value and the unit of measurement, and in others, no space is left.

PLOS authors have the option to publish the peer review history of their article (what does this mean? ). If published, this will include your full peer review and any attached files.

**Do you want your identity to be public for this peer review?** For information about this choice, including consent withdrawal, please see our Privacy Policy .

Reviewer #1: No

Reviewer #2: No

Reviewer #3: **Yes: ** Ahmed S. Attia

**Figure resubmission:**

**Reproducibility:**



---

## [Decision Letter · Decision Letter 1]

20 Aug 2025

Dear Dr. St. Geme III,

We are pleased to inform you that your manuscript 'Identification of a Kingella kingae factor H binding protein that is the major determinant of serum resistance' has been provisionally accepted for publication in PLOS Pathogens.

Before your manuscript can be formally accepted you will need to complete some formatting changes, which you will receive in a follow up email. A member of our team will be in touch with a set of requests. Please include the modification asked by #R2.

Best regards,

Mathieu Coureuil

Guest Editor

PLOS Pathogens

Matthew Wolfgang

Section Editor

PLOS Pathogens

Sumita Bhaduri-McIntosh

Editor-in-Chief

PLOS Pathogens

orcid.org/0000-0003-2946-9497

Michael Malim

Editor-in-Chief

PLOS Pathogens

orcid.org/0000-0002-7699-2064

Editor Comments:

Please modify the label of the Y-Axis in Fig. 7B as asked by #R2 and send back the manuscript.

Reviewer Comments (if any, and for reference):

Reviewer's Responses to Questions

**Part I - Summary**

Reviewer #1: The authors have satisfactorily modified the manuscript according to the reviewers’ comments.

The article is now ready for final acceptance and publication.

Reviewer #2: The authors have provided a very thoughtful and thorough response to my critiques. Only one minor editorial issue related the the label of the Y-Axis in Fig. 7B as mentioned below needs to be fixed. This is a solid paper that defines an importance virulence factor of K. kingae.

Reviewer #3: The authors have addressed my comments on the original manuscript in a satisfactory manner.

**Part II – Major Issues: Key Experiments Required for Acceptance**

Reviewer #1: The authors have satisfactorily modified the manuscript according to the reviewers’ comments.

The article is now ready for final acceptance and publication.

Reviewer #2: None

Reviewer #3: The authors have addressed my comments on the original manuscript in a satisfactory manner.

**Part III – Minor Issues: Editorial and Data Presentation Modifications**

Reviewer #1: The authors have satisfactorily modified the manuscript according to the reviewers’ comments.

The article is now ready for final acceptance and publication.

Reviewer #2: Please change Y-axis label on Fig 7B to NHS-Mg-EGTA/HI-NHS (not NHS/HI-NHS, which is the same as Fig. 7A).

Reviewer #3: The authors have addressed my comments on the original manuscript in a satisfactory manner.

PLOS authors have the option to publish the peer review history of their article (what does this mean? ). If published, this will include your full peer review and any attached files.

**Do you want your identity to be public for this peer review?** For information about this choice, including consent withdrawal, please see our Privacy Policy .

Reviewer #1: No

Reviewer #2: No

Reviewer #3: **Yes: ** Ahmed S. Attia

---

## [Editor Report · Acceptance letter]

Dear Dr. St. Geme III,

We are delighted to inform you that your manuscript, " 

Identification of a Kingella kingae factor H binding protein that is the major determinant of serum resistance," has been formally accepted for publication in PLOS Pathogens.

Best regards,

Sumita Bhaduri-McIntosh

Editor-in-Chief

PLOS Pathogens

orcid.org/0000-0003-2946-9497

Michael Malim

Editor-in-Chief

PLOS Pathogens

orcid.org/0000-0002-7699-2064